

# Information transfer with a gravitating bath

**Hao Geng[1⋆], Andreas Karch[1,2†], Carlos Perez-Pardavila[2‡],
Suvrat Raju[3∘], Lisa Randall[4§], Marcos Riojas[2¶] and Sanjit Shashi[2‖]**

**1** Department of Physics, University of Washington, Seattle, WA, 98195-1560, USA.
**2** Theory Group, Department of Physics, University of Texas, Austin, TX 78712, USA.
**3** International Centre for Theoretical Sciences, Tata Institute of Fundamental Research,
Shivakote, Bengaluru 560089, India.
**4** Harvard University, 17 Oxford St., Cambridge, MA, 02139, USA.

⋆ hg666@uw.edu, † karcha@utexas.edu, ‡ cjp3247@utexas.edu,
∘ suvrat@icts.res.in, § randall@g.harvard.edu,
¶ marcos.riojas@utexas.edu, ‖ sshashi@utexas.edu

## Abstract

Late-time dominance of entanglement islands plays a critical role in addressing the information paradox for black holes in AdS coupled to an asymptotic non-gravitational bath. A natural question is how this observation can be extended to gravitational systems. To gain insight into this question, we explore how this story is modified within the context of Karch-Randall braneworlds when we allow the asymptotic bath to couple to dynamical gravity. We find that because of the inability to separate degrees of freedom by spatial location when defining the radiation region, the entanglement entropy of radiation emitted into the bath is a time-independent constant, consistent with recent work on black hole information in asymptotically flat space. If we instead consider an entanglement entropy between two sectors of a specific division of the Hilbert space, we then find non-trivial time-dependence, with the Page time a monotonically decreasing function of the brane angle—provided both branes are below a particular angle. However, the properties of the entropy depend discontinuously on this angle, which is the first example of such discontinuous behavior for an AdS brane in AdS space.



# 1 Introduction

An important and interesting advance in our understanding of the quantum mechanics of black holes is the set of recent calculations [1, 2] of the time evolution of the entanglement entropy between a holographic system that contains a black hole and an external bath. (See [3–5] for recent reviews). These calculations have yielded the so-called *Page curve* for the time-dependence of the entanglement entropy, which is consistent with unitarity and general expectations from quantum information theory [6,7]. One crucial ingredient in this calculation is the appearance of *entanglement islands* [8–11], which are seemingly disconnected parts in the holographic system that contribute to the entanglement entropy of a region of the bath.

Attempts have been made to generalize this calculation to higher-dimensional models as well [9, 12–16]. A common feature of most such models to date is that the external bath into which the black hole evaporates is non-gravitating. It was recently pointed out in [12] that in all such models with a propagating graviton, this graviton is massive as a direct consequence of coupling the gravitational theory to the bath. Take away the coupling, then you take away the mass. To what extent this coupling to the bath, and hence the mass of the graviton, is crucial in these calculations remains to be seen.[1]

Using somewhat orthogonal methods, the time evolution of the entanglement entropy of black holes in asymptotically flat space was recently studied in [20]. One of the claims of [20] is that for black holes in asymptotically flat space, all information about the exact quantum state of the full spacetime (including the black hole) is accessible from an infinitesimal neighborhood of the past boundary of future null infinity. If true, the entanglement between the black hole degrees of freedom and the radiation reaching null infinity is completely time-independent, and the entropy curve would be flat instead of first rising and later falling.

In order to compare and contrast this result with the results of [1–3], the authors of [20] noted that the constructions of [1–3] have a non-gravitating external bath, so local observables

---

[1]One recent paper that explores the construction of entanglement islands and Page curves for black hole radiation in a massless theory of gravity, and where the radiation region is also gravitating, is [13]. It deserves to be mentioned that we can construct entanglement islands in a massless theory of gravity without any information paradoxes to start with [17–19].

in the bath in these calculations are well-defined, as is always true in quantum field theory. Since the radiation region in this scenario is taken to be part of the non-gravitating bath, the Hilbert space can be factorized into a part that describes the radiation and its complement. In contrast, gravity in asymptotically flat space does not shut off near null infinity. So it was suggested by [20] that, in order to mimic this situation and see if the Page curve calculations persist in gravitating systems, one would need to at least weakly couple the external bath to gravity. However, even the presence of weak gravity can potentially change the answers to fine-grained quantum information questions as was recently emphasized in [21].

Most work on islands in higher dimensions [9,12–16,22–26], as well as some of the results in lower dimensions [27–32], have been obtained in the context of the *Randall-Sundrum (RS)* braneworlds [33] with subcritical tension branes [34, 35], also known as the *Karch-Randall (KR)* braneworlds. Compared to the generic RS setup, KR has several specific features. It allows not just two but in fact three different holographically equivalent descriptions. Furthermore, gravity is generically massive, with an almost massless graviton only emerging in the near flat limit.

It is worth noting that, like the original calculation of [6], previous calculations also assumed a non-gravitating bath whereas we seem to live in a world with gravitating degrees of freedom. So in this paper we consider another important question allowed by this type of set-up: how the entanglement entropy calculation would generalize to a gravitating bath.

As with any RS scenario, the KR setup has the nice property that it is straightforward to introduce weak gravity in the bath region: one simply includes a second KR brane. The resulting setup is sketched in Figure 1a. A critical property of this generalization is that in this double KR branes setup, in contrast to the single brane case discussed in [12], the model contains a massless graviton since the warped extra dimension now has a finite volume. The light (but massive) graviton that appears in the limit where the original brane is near the boundary coexists with the truly massless graviton—a phenomenon that was dubbed bi-gravity in [36]. The massless graviton is a superposition of the modes localized near the left brane and the right brane. In the limit that one brane, designated as the bath, is weakly gravitating (in the sense that it's closer to the boundary), the massless graviton is mostly localized on this weakly gravitating "bath" brane but also has some small overlap with the strongly gravitating "physical" brane. Together with the arguments put forward in [20] this seems to be an ideal scenario to investigate the significance of entanglement islands for black hole radiation in a system with gravitating bath.

Based on these motivations and observations, we will study two distinct types of entanglement entropy for our system with a gravitating bath: one closer to previous work on the entropy of the black hole radiation that we demonstrate has no time-dependent Page curve and another entropy that we define which follows an interesting Page curve even in this system. In both cases we consider a zero temperature (vacuum) setup as well as one with a thermal black string configuration [37, 38]. The latter is perhaps the simplest and most readily calculable way to induce a black hole on the brane.[2] We consider eternal, $(d + 1)$-dimensional geometries. Our numerical results primarily focus on $d = 4$.

Our first approach mimics earlier calculations in that we study the entanglement entropy of a radiation region on the bath brane. A key difference between a gravitating bath and a non-gravitating bath is that in the former, there is no natural, diffeomorphism-invariant way to separate local degrees of freedom into different regions, so the radiation region $\mathcal{R}$ on the bath brane should be determined *dynamically*. We suggest a modest extension of previous dynamical principles to determine $\mathcal{R}$ for KR two-brane setups by studying *quantum extremal surfaces (QES)* [39] via a doubly-holographic model, for which the location and the associated entropy of the QES in the $d$-dimensional theory is determined by a classical minimal (RT)

---

[2]Though both branes contain a black hole, we divide the setup by fiat into a physical and bath region.

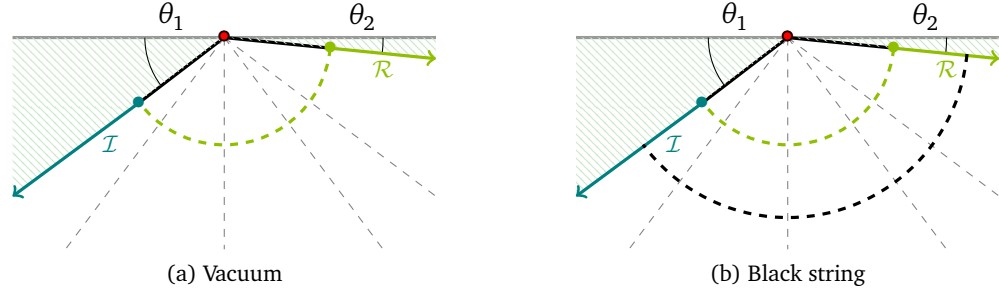

Figure 1: (a) Embedding of a KR braneworld with two subcritical branes in anti-de
Sitter space. We refer to the left brane as the "physical" brane, and the right brane
as the "bath" brane. $\mathcal{R}$ denotes the radiation region whose entanglement entropy we
wish to calculate, and the dashed green line connected to the boundary of $\mathcal{R}$ is the
candidate RT surface. $\mathcal{I}$ is the entanglement island on the brane corresponding to the
RT surface in green. (b) Embedding of a KR braneworld with the same tensions in the
black string geometry. The dashed black line is the black string horizon separating
the exterior and interior regions.

surface in a $(d + 1)$-dimensional bulk.

Following this procedure, we will see that in the zero temperature case (global AdS), $\mathcal{R}$ on the bath brane reduces to a point, and the entanglement entropy is exactly zero, which is consistent with the setup being dual to a pure state. However, for the black string geometry, we find that the RT surface is the horizon and $\mathcal{R}$ is the region behind the horizon, and the entanglement entropy is given by the area of the horizon, *i.e.* the Bekenstein-Hawking entropy. Regardless, for both configurations we reach an agreement with arguments in [20], in the sense that, as the entanglement surface remains constant in time, the corresponding entropy is also constant.

In fact, [20] also suggested that while the Page curve may not describe the fine-grained entropy of the radiation when gravity is dynamical, there are *other questions* to which the Page curve might still be the answer. And, indeed, this is exactly what we find in this paper.

There is a second kind of question we can ask in our setup. Gravity switches off completely on the $(d-1)$-dimensional boundary defect where both branes meet. (This point is marked as a red dot in both Figure 1a and Figure 1b). Therefore, it is again possible to factorize the Hilbert space on this boundary. This factorization allows us to perform an analysis that is analogous to those of [1–3]. We can divide the boundary Hilbert space into different subsectors and ask about the entanglment entropy between them. We refer to the resulting quantity as *left/right entanglement entropy*. In the absence of entanglement islands this would correspond to entanglement between the left and right branes, although we explain below that this interpretation must be modified in the presence of islands.

We will place our results in the context of both older [40] and more recent [41,42] progress on such systems, in which a $(d-1)$-dimensional CFT is dual to a $(d+1)$-dimensional gravitational space. In the more recent papers, this has been termed *wedge holography*. The left/right division alluded to above, refers to a well-defined *internal division* of the degrees of freedom of this $(d-1)$-dimensional CFT into two subsystems. These subsystems are coupled but correspond to distinct factors of the Hilbert space, and so one can ask about the entanglement entropy between them and also the time-dependence of this entanglement entropy.

The left/right entanglement entropy is calculated in the $(d+1)$-dimensional bulk as the smallest of two classes of candidate minimal surfaces anchored to the defect:

- the *Hartman-Maldacena surface* [43], which stretches from the defect, where the two

branes meet on the asymptotic conformal boundary, to its thermofield double partner, and

- the minimal surface starting from the defect and shooting towards one of the branes.

This second class of surfaces tells us that sometimes degrees of freedom on the right brane may be described by degrees of freedom from the left sector of the $(d-1)$-dimensional CFT. This is why the left/right entanglement entropy should not be interpreted as the entanglement entropy between the left and the right brane.

We also show that in the black string geometry, the Hartman-Maldacena surface must cross an Einstein-Rosen (ER) bridge, so it grows in time, with the growth being linear at late times. Put another way, because the geometry has a horizon and is a fast scrambler [44], the growth of this surface is indefinite.[3] As the other candidate surface is constant in time, the Hartman-Maldacena surface should, at some point, be larger in area, allowing for the possibility of a phase transition for the entangling surface.

This is exactly the recent story of entanglement islands in the KR braneworld that gives a Page curve resolving a version of the information paradox for the eternal black hole [8]. We study the left/right entanglement entropy in empty $AdS_{d+1}$ and the $AdS_{d+1}$ black string. Surprisingly, we find that the time-dependent Page curve for this left/right entropy appears if and only if the two branes are below a particular angle called the *Page angle*, $\theta_P$.

In our analysis, we find that above an angle slightly larger than the Page angle, called the *critical angle*, $\theta_c$, the island surfaces cease to exist altogether. The entropy is then governed by a limiting surface that we call the *tiny island surface* and describe in more detail in section 3.2.3. The value of $\theta_c$, as we will find in our $d = 4$ numerics, is independent of the specifics of the bulk geometry *except* for the dimension because this value is controlled by the asymptotic region near the AdS boundary. We use this property to derive an analytical formula for $\theta_c$ using empty $AdS_{d+1}$.

Our paper is organized as follows. In Section 2, we explore the entanglement entropy of dynamical radiation regions in empty $AdS_{d+1}$ and the black string geometry with two branes. In Section 3, we study the left/right entanglement entropy of the black string, for which we may find a Page curve. In Section 4, we discuss the apparent universality of the critical angle $\theta_c$ which controls the appearance of the Page curve. We compute $\theta_c$ for general $d$ using empty $AdS_{d+1}$ and examine some of its properties. Finally, in Section 5, we conclude with additional discussions and remarks guiding future directions of interest. Appendix A provides details of the area growth of the Hartman-Maldacena surface in the black string geometry.

## 2 Entanglement Entropy of a Dynamical $\mathcal{R}$

In this section, we study the entanglement entropy of a radiation region $\mathcal{R}$ on the bath brane which, as we will see in the following examples, can be redundant with degrees of freedom on the physical brane. We consider both zero temperature and finite temperature geometries.

In our zero temperature example, the bulk geometry consists of two $AdS_d$ branes in empty $AdS_{d+1}$, both of which have positive tension. At finite temperature, we take the bulk geometry to be the $AdS_{d+1}$ black string. In this case the induced geometries on both branes will be $AdS_d$ planar black holes.

We argue that $\mathcal{R}$ should be dynamically determined by the extremization of the entangling surface, as with previous quantum extremal surface prescriptions in the gravitating region,

---

[3]A similar statement would be true to other types of geometries with event horizons, such as de Sitter space which was studied in [45–47].

with important consequences for the resultant entanglement entropy. In Section 2.1, we encapsulate the motivation for the dynamical extremization principle to determine $\mathcal{R}$. In Section 2.2, we show that in the context of Karch-Randall braneworlds, the extremization procedure specifies that the entangling surface is perpendicular to both branes (Neumann boundary conditions). We emphasize that the entanglement entropy of $\mathcal{R}$ is constant in time for both cases discussed in Sections 2.3 and 2.4, as anticipated by the analysis in [20].

## 2.1 Double Holography and Quantum Extremal Surfaces

In this subsection, we will review how KR braneworlds can be naturally understood as *doubly-holographic* and elaborate how holography helps us calculate entanglement entropy. This holographic dictionary encodes the von Neumann entropy of the black hole radiation in a classical minimal surface residing in a higher-dimensional gravity theory, with the boundary of this surface setting the location of the entanglement island. The ultimate purpose of this section is to propose an extension of this holographic dictionary from the usual one-brane to a two-brane setup. We will see that because the radiation degrees of freedom whose entropy we calculate are also located on a gravitating region, the minimization of the bulk classical surface serves as a dynamical principle which decides not just the location of the island, but also the location of the radiation region.

Let us first briefly review the story of a single KR brane coupled to a non-gravitating bath and its three descriptions. We call such a system doubly-holographic because these three descriptions are related to each other by applying the standard AdS/CFT holography twice [27, 34, 35]:

(I) a $d$-dimensional CFT on a flat background with a $(d-1)$-dimensional boundary (i.e. a BCFT$_d$ [48, 49]), a description also emphasized in [50, 51],

(II) a $d$-dimensional CFT with some characteristic UV cutoff coupled to gravity on an asymptotically AdS$_d$ space[4] $\mathcal{M}_d$, with the boundary of $\mathcal{M}_d$ connected to another $d$-dimensional CFT on a half-Minkowski space via transparent boundary conditions,

(III) Einstein gravity in an asymptotically AdS$_{d+1}$ space $\mathcal{M}'_{d+1}$ containing a Karch-Randall brane [34] $\mathcal{M}_d$ as an end-of-the-world brane.

In keeping with previous calculations with a non-gravitating bath, we first consider the von Neumann entropy $S(\mathcal{R})$ of a codimension-one (i.e. on a constant time slice) subregion $\mathcal{R}$ defined in the description (I). In the description (II), using the quantum corrected *Ryu-Takayanagi formula* [52–54], this is given by the so-called *island rule* [1, 27] as,

$$S(\mathcal{R}) = \min_{\mathcal{I}} \text{ext}\, S_{\text{gen}}(\mathcal{R} \cup \mathcal{I}), \tag{1}$$

where $\mathcal{I}$ is a codimension-one region in $\mathcal{M}_d$ disconnected from $\mathcal{R}$ and thus called the *island*. $S_{\text{gen}}$ denotes the *generalized entropy* functional used in the quantum extremal surface (QES) prescription of [39],

$$S_{\text{gen}}(\mathcal{R} \cup \mathcal{I}) = \frac{A(\partial \mathcal{I})}{4G_N} + S_{\text{matter}}(\mathcal{R} \cup \mathcal{I}), \tag{2}$$

and $\partial \mathcal{I}$ is the QES. Here we want to emphasize that $\mathcal{I}$ lives on the gravitating region and $\mathcal{R}$ is on a non-gravitating region. Therefore, in (1) we minimize $S_{\text{gen}}$ over $\partial \mathcal{I}$ to obtain the von Neumann entropy, and this minimization in the gravitating region can be thought of as a quantum manifestation of the diffeomorphism invariance. The area term comes from using

---

[4]For the minimal scenario, this is just AdS$_d$ itself.

holography to promote the $(d-1)$-dimensional degrees of freedom in the BCFT$_d$ of description (I) to gravitational degrees of freedom on $\mathcal{M}_d$, yielding (II).

$G_N$ is the Newton's constant on $\mathcal{M}_d$. In the context of RS braneworlds, gravity can be thought of as entirely induced by matter loops after removing UV degrees of freedom. In this interpretation $G_N^{-1}$ vanishes at tree level and is induced only by matter loops. Without a brane, this induced $G_N^{-1}$ would be divergent, but since on the brane the CFT$_d$ has a cutoff, this induced contribution is finite. In practice, this means that the area term in the special case of induced gravity is included in $S_{\text{matter}}$ as long as no explicit Ricci scalar term[5] is included in the brane action [16, 56].

Using holography a second time helps us compute $S_{\text{matter}}$, which corresponds to the CFT degrees of freedom on the $d$-dimensional configuration of description (II). Specifically, the CFT matter in (II) is promoted to bulk gravity in (III), so to the leading semiclassical order $S_{\text{matter}}$ can be computed using only the geometrical data in $\mathcal{M}'_{d+1}$. Since gravity on the brane is induced only by matter effects, $S_{\text{gen}}$ itself is encoded by the geometry of $\mathcal{M}'_{d+1}$ in accordance with the standard Ryu-Takayanagi prescription (assuming that $\mathcal{R}$ is on a constant-time slice).

Therefore, we can write the resulting generalized entropy in terms of the area of a $(d-1)$-dimension surface $\gamma$ cutting through the $(d+1)$-dimensional spacetime from $\partial\mathcal{R}$ to $\partial\mathcal{I}$,

$$S_{\text{gen}}(\mathcal{R} \cup \mathcal{I}) = \frac{A(\gamma)}{4G_{d+1}}, \tag{3}$$

where $G_{d+1}$ is the Newton's constant of $\mathcal{M}'_{d+1}$ in the description (III).

Combining this with (1), we get the doubly-holographic rule to compute $S(\mathcal{R})$—we minimize $S_{\text{gen}}$ by varying $\mathcal{I}$ and thus the position of $\partial\mathcal{I}$. This dynamical principle is essential because $\mathcal{I}$ is defined in a gravitating region. However, $\mathcal{R}$ is fixed beforehand, since the system described by (I) is non-gravitating and thus can be partitioned.

Now we may ask what happens when the bath is gravitating as well. We still have a doubly-holographic set-up, but involving the *wedge holography* discussed by [41, 42]:

(I*) a $(d-1)$-dimensional CFT on $\mathcal{M}^{(0)}_{d-1}$. Here the BCFT is replaced by $(d-1)$-dimensional surface since both branes have a lower-dimensional holographic dual.

(II*) two $d$-dimensional CFTs with characteristic UV cutoffs coupled to gravity on distinct asymptotically AdS$_d$ spaces $\mathcal{M}^L_d$ and $\mathcal{M}^R_d$, with these systems being connected via transparent boundary conditions at a defect $\mathcal{M}^{(0)}_{d-1} = \partial\mathcal{M}^L_d = \partial\mathcal{M}^R_d$,

(III*) Einstein gravity in an asymptotically AdS$_{d+1}$ space $\mathcal{M}'_{d+1}$ containing two end-of-the-world Karch-Randall branes $\mathcal{M}^{(L)}_d$ and $\mathcal{M}^{(R)}_d$, both of which are gravitational regions and both of which are attached at a defect and thus form a *wedge*.

However, as there is no fundamental distinction between the physical and the bath branes, now the quantum manifestation of the diffeomorphism invariance should be a straightforward generalization of the island rule to demand that instead of the single brane instructions in (1) and (2), we extremize $S_{\text{gen}}$ over *both* $\partial\mathcal{I}$ and $\partial\mathcal{R}$:

$$S = \min_{\mathcal{I},\mathcal{R}} \text{ext}\, S_{\text{gen}}(\mathcal{R} \cup \mathcal{I}), \tag{4}$$

$$S_{\text{gen}}(\mathcal{R} \cup \mathcal{I}) = S_{\text{matter}}(\mathcal{R} \cup \mathcal{I}). \tag{5}$$

As we mentioned above, in the context of RS braneworlds, where gravity is entirely induced, the generalized entropy is determined by matter effects alone so we don't include

---

[5] [27] and [15, 16] included a *Dvali-Gabadadze-Porrati (DGP) gravity* [55] term in the brane action, that is an explicit term proportional to the Ricci scalar of the induced metric. This is a higher curvature term in the brane's action. In this case, more care is needed beyond simply having induced RS gravity [16].

separate area terms. Now using holography, we know that $S_{\text{matter}}(\mathcal{R} \cup \mathcal{I})$ can be calculated in terms of a classical RT surface $\gamma$ connecting $\partial\mathcal{R}$ to $\partial\mathcal{I}$ in the higher-dimensional spacetime $\mathcal{M}'_{d+1}$

$$S = \min_{\mathcal{I},\mathcal{R}} \text{ext} \frac{A(\gamma)}{4G_{d+1}}. \tag{6}$$

Our proposal (6) yields both an island $\mathcal{I}$ and a radiation region $\mathcal{R}$ whose boundaries are chosen to minimize the area of the surface $\gamma$. That we have no inputs at all is a result of the entire system being gravitating—we simply cannot choose a radiation region because this would involve partitioning gravitational degrees of freedom. The extremization over $\gamma$ also includes extremizing over its endpoints, which is the equivalent of extremizing over $\mathcal{R}$ and $\mathcal{I}$. This minimization over the boundary of $\gamma$ eventually translates to boundary conditions of $\gamma$ near the two branes. Since these boundary conditions are dynamically determined, we will call them dynamical boundary conditions.

We emphasize that there are several physical arguments one can make to support this claim about boundary conditions. First of all, with both branes gravitating, it seems unjustified to treat them asymmetrically as we would do by imposing a fixed radiation region as a boundary condition only on the physical brane. Second, in a theory with gravity, it is not meaningful to localize degrees of freedom in a region that does not extend to asymptotic infinity. This is in accordance with the well-known result that there are no local gauge-invariant observables in quantum gravity [57–59]. Furthermore, allowing for dynamical boundary conditions also appears to be what is required by the philosophy of quantum extremal surfaces [39]; in a theory with dynamical gravity, the position of the surface across which we calculate the entanglement entropy of the matter fields[6] is not fixed a priori, but needs to be extremized over.

Our procedure is distinct from [60] and also from [61], where the authors attempted to define the entropy of a region with dynamical gravity using a "relational" procedure. In our context, the analog of this procedure would be to identify points in the radiation region by shooting geodesics from the defect and following them for a given proper distance along the brane. However, note that this procedure does not actually divide the algebra of operators on the brane into two commuting subalgebras. The geodesic that connects the operator in the radiation region to the defect can be thought of as a gravitational Wilson line that extends *outside* the radiation region. Since every operator that appears to be localized in the radiation region is actually attached to such a Wilson line, it does not commute with operators outside the region. The notion of a fixed radiation region may still be useful to define a more coarse-grained entropy [62], where one somehow sidesteps these gravitational effects, but not to compute a fine-grained entropy, as we are doing here. This issue was also recently discussed in [63], where it was emphasized that some observables may need to be systematically discarded from the algebra to define the entanglement-entropy of a fixed region in gravity.

## 2.2 Extremization and Neumann Boundary Conditions

The geometries we wish to consider are depicted in Figure 1. The physical brane on the left is at an angle $\theta_1$, the bath brane on the right at $\theta_2$. We will generally be interested in scenarios in which $\theta_2 \ll \theta_1$ so that the bath is weakly gravitating when compared to the physical brane, but most results will apply for general angles. Both branes are taken to be subcritical, so the geometry of the weakly gravitating bath is also asymptotically $\text{AdS}_d$, with a curvature radius that is much larger than that of the physical asymptotically $\text{AdS}_d$ brane when the bath brane is at a small angle. When we take much smaller $\theta_2$, the truly massless graviton mostly overlaps

---

[6]In our doubly-holographic model, recall that the $(d + 1)$th bulk dimension may be viewed as matter on the branes.

with the bath brane, but also has an exponentially suppressed overlap with the physical brane. The radiation region $\mathcal{R}$ is a subregion of the bath, indicated by a solid green line in Figure 1a.

In establishing some technical details and obtaining the boundary conditions determining $\mathcal{R}$, we first consider the vacuum solution. While islands are of particular importance when studying black holes on the physical brane (potentially in thermal equilibrium with the bath if the latter is kept at the same finite temperature as the black hole), it was recently shown in [16] that a lot can be learned already about the existence of islands from studying the zero temperature case.

In order to describe the system quantitatively, we need to commit to a coordinate system. There are two useful coordinate systems. One is the standard *Poincaré patch* coordinate system on $\text{AdS}_{d+1}$ (setting the bulk curvature radius to 1),

$$ds^2 = \frac{1}{z^2}(-dt^2 + dz^2 + d\vec{x}^2 + dy^2). \tag{7}$$

Here $y \in \mathbb{R}$ is the horizontal direction in Figure 1, $z > 0$ is the vertical direction, and $\vec{x} = (x^1, ..., x^{d-2})$ represents the $d-2$ real transverse directions which are suppressed in the figure together with $t \in \mathbb{R}$. Here the boundary of $\text{AdS}_{d+1}$ is located at $z \to 0$, and the brane ends on this boundary at $y = 0$.

The other coordinate system is the one adapted to the geometry of the subcritical KR brane,

$$ds^2 = \frac{1}{\sin^2 \mu} \left( \frac{-dt^2 + du^2 + d\vec{x}^2}{u^2} + d\mu^2 \right). \tag{8}$$

In Figure 1, $u > 0$ and $\mu \in (0, \pi)$ are radial and angular coordinates of a *spherical* coordinate system centered on the defect at which the two branes meet. In these coordinates, each subcritical brane is located at a constant $\mu$. For example, the two branes in Figure 1a are located at $\mu = \theta_1$ and $\mu = \pi - \theta_2$. Several other lines of constant $\mu$ are indicated as dashed lines. The geometry of a slice with constant $\mu$ is itself $\text{AdS}_d$, with $u$ being the radial Poincaré patch coordinate on each slice, *i.e.* the brane analog of $z$. The change of coordinates from $y$-$z$ to $u$-$\mu$ is given by,[7]

$$z = u \sin \mu, \quad y = -u \cos \mu. \tag{9}$$

Let us work out the minimal area surfaces connecting to a brane in this spacetime. To find the equations of motion of the bulk minimal surface, it is easiest to work with the coordinates in (7). In this coordinate system, the area density per unit volume[8] in the transverse space labelled by $\vec{x}$ for an embedding $y(z)$ can we written as,

$$\mathcal{A} = \int \frac{dz}{z^{d-1}} \sqrt{1 + y'(z)^2}. \tag{10}$$

Since $y(z)$ appears only in the "action" via its derivative, we can straightforwardly integrate the Euler-Lagrange equation to,

$$\frac{1}{z^{d-1}} \frac{y'(z)}{\sqrt{1 + y'(z)^2}} = \pm \frac{1}{z_*^{d-1}} \implies y'(z) = \pm \frac{z^{d-1}}{\sqrt{z_*^{2(d-1)} - z^{2(d-1)}}}. \tag{11}$$

Here, the integration constant $z_* > 0$ denotes the depth of the *turnaround point* of the minimal surface. The minimal area surface reaches towards the boundary (meaning to smaller

---

[7]The sign here is chosen so that $y$ is negative when $\mu$ is between 0 and $\pi/2$ and $y$ is positive when $\mu$ is between $\pi/2$ and $\pi$.

[8]The transverse volume will play no role in our work. Strictly speaking all entropies we calculate are entropy densities per unit transverse volume.

$z$) at both sides of the turnaround point. At $z_*$ the two branches of the solution get glued together. Note that the *only* solution which does not have a turnaround point is $y' = 0$, the straight vertical line corresponding to $z_* \to \infty$. This is the only solution that passes through the Poincaré patch horizon. All other solutions will turn around and so reach either one of the two branes, hence giving rise to an island ($\mathcal{I}$ in Figure 1a).

The novel aspect of our calculation involves the boundary condition imposed on the bath brane. To determine the correct boundary conditions on the brane, it is easier to work in the $u$-$\mu$ coordinates. We consider the case where the actions for the branes contain only their tension, as is natural from the point of view of a low energy effective action. The tension term is leading in a derivative expansion. Thus in this case, there is no contribution to the entanglement entropy from the brane action, and as we discussed in Sec.2.1, the correct boundary conditions simply arise from minimizing the area functional over all possible locations of the of the surface along the boundary.

In this setup, the locations where the RT surface meets the branes are allowed to vary, and we minimize the area of the surface with respect to this location. Let us first review this analysis. In $u$-$\mu$ coordinates, our action for an embedding $u(\mu)$ reads,

$$\mathcal{A} = \int_{\theta_1}^{\pi-\theta_2} \frac{d\mu}{(u\sin\mu)^{d-1}} \sqrt{u^2 + u'(\mu)^2}. \tag{12}$$

Note that the integration boundaries are important in specifying the boundary conditions. The action contains both $u$ and $u'$, and so the equations of motion appear complicated, even though we know that the most general solution still must be given by (11) processed via the coordinate change (9).

When deriving the equations of motion we need to integrate by parts. Keeping track of boundary terms, this means,

$$0 = \delta\mathcal{A} = \int_{\theta_1}^{\pi-\theta_2} \frac{d\mu}{(u\sin\mu)^{d-1}} \frac{u'}{\sqrt{u^2 + u'^2}} \delta u' + \int_{\theta_1}^{\pi-\theta_2} \frac{\delta A}{\delta u} \delta u \, d\mu$$

$$= \frac{\delta u}{(u\sin\mu)^{d-1}} \frac{u'}{\sqrt{u^2 + u'^2}} \Big|_{\theta_1}^{\pi-\theta_2} - \int_{\theta_1}^{\pi-\theta_2} d\mu \, (\text{EOM}) \delta u. \tag{13}$$

The second term is the bulk equation of motion. While non-trivial in these coordinates, we know that the most general solution is given by (11) above. The first term is the boundary term obtained by integrating by parts. For it to vanish, we impose the following Neumann boundary conditions,

$$u'(\theta_1) = u'(\pi - \theta_2) = 0. \tag{14}$$

That is, the RT surface ends orthogonally to both branes. Note that this is the simplest type of boundary condition we may take which does not involve fixing a point on the brane and thus partitioning a gravitational system. Using the change of coordinates (9), we can readily see what this implies for a curve $y(z)$ in the $y$-$z$ coordinate system.

$$\frac{dz}{d\mu} = u'\sin\mu + u\cos\mu, \quad \frac{dy}{d\mu} = -u'\cos\mu + u\sin\mu, \tag{15}$$

and hence,

$$\frac{dy}{dz} = \frac{dy/d\mu}{dz/d\mu} = \frac{u\sin\mu - u'\cos\mu}{u'\sin\mu + u\cos\mu}. \tag{16}$$

Using (14), this implies that on the brane

$$y' = \tan\mu|_{\mu=\theta_1,\pi-\theta_2} = -\frac{z}{y}. \tag{17}$$

This is how orthogonality reads in the $y$-$z$ coordinates. How could we have seen this boundary condition directly from the $y$-$z$ coordinates? The problem is that, in the action (10), varying the endpoint of the RT surfaces along the brane does not just change the variable $y$ of the endpoint, but also the $z$-value that appears as a limit of the integration region. To deal with this complication, it is easiest to parameterize the RT surface as $(y(\tau), z(\tau))$, where $\tau$ runs from 0 to 1. With this, the $y$-$z$ action reads,

$$\mathcal{A} = \int_0^1 d\tau \frac{\sqrt{\dot{y}^2 + \dot{z}^2}}{z^{d-1}}, \tag{18}$$

where dots represent differentiation with respect to $\tau$.

This time, we see that the boundary term we pick up from integrating by parts in deriving the equations of motions demands,

$$\dot{y}(\delta y) + \dot{z}(\delta z)|_{\theta_1} = \dot{y}(\delta y) + \dot{z}(\delta z)|_{\pi - \theta_2} = 0. \tag{19}$$

To see what this means for $y(z)$, note that the boundary variation of $y$ and $z$ aren't independent. Let us focus on the brane at $\mu = \theta_1$; the boundary condition at $\mu = \pi - \theta_2$ is analogous. For the boundary of the RT surface to lie on the brane, whose embedding is given by the equation $z = -y \tan\theta_1$, we need to require,

$$\delta z = -(\delta y) \tan\theta_1. \tag{20}$$

This means that our boundary condition is,

$$\frac{dy}{dz} = \frac{\dot{y}}{\dot{z}} = -\frac{\delta z}{\delta y} = \tan\theta_1, \tag{21}$$

in perfect agreement with (17). We find a similar expression at the $\mu = \pi - \theta_2$ brane.

We will also be studying the $AdS_{d+1}$ *black string*,

$$ds^2 = \frac{1}{u^2 \sin^2\mu} \left[ -h(u)dt^2 + \frac{du^2}{h(u)} + d\vec{x}^2 + u^2 d\mu^2 \right], \quad h(u) = 1 - \frac{u^{d-1}}{u_h^{d-1}}. \tag{22}$$

These coordinates are the same as those of $AdS_{d+1}$ in spherical coordinates, so $u > 0$ and $\theta_1 \le \mu \le \pi - \theta_2$ with the branes present. The action for an embedding $u(\mu)$ which is analogous to (12) is,

$$\mathcal{A} = \int_{\theta_1}^{\pi - \theta_2} \frac{d\mu}{(u \sin\mu)^{d-1}} \sqrt{u^2 + \frac{u'(\mu)^2}{h(u)}}. \tag{23}$$

The equations of motion are more complicated because of the blackening factor $h(u)$, but the boundary conditions at the branes are still those of empty $AdS_{d+1}$ (14).

For a non-gravitating bath, one demands Dirichlet boundary conditions, *i.e.* fixing a point $y_0$ as the endpoint of the RT surface on the bath so that $y_0$ defines the radiation region of interest. Fixing the endpoint would also satisfy the general boundary condition in (13). However, as we discussed in Section 2.1, for a gravitating bath, the correct prescription is to minimize over all possible endpoints of the RT surface on the bath brane. We saw in this section that this is translated into imposing the Neumann condition (14) on the bath brane. So instead of working with a fixed radiation region $\mathcal{R}$, we let the location of $\partial\mathcal{R}$ adjust in order to minimize the entropy. Note that this doesn't contradict with the Dirichlet boundary condition if we continuously dial the bath brane to be a non-gravitating bath, because any extremal surface will automatically satisfy $y' = 0$ as $z \to 0$, according to (11).

The procedure of imposing this boundary condition on the brane has an interesting implication. Immediately, we see that the trivial surface, *i.e.* the $y' = 0$ surface such as that of

Hartman and Maldacena [43], is no longer allowed. As mentioned above, the only island-free RT surface, that is the only surface that does not turn around, is this trivial surface. But it can never be orthogonal to the bath brane if $\theta_2 \neq 0$. Thus, we have a unique RT surface whose area doesn't grow in time, and whose detailed form we will work out below, which always dominates. Hence, the entropy curve is flat, as in [20].

We conclude this subsection by linking our work to the analysis of [20] and the wedge holography of [41, 42]. In asymptotically Minkowski space, gravity is dynamical everywhere. As a result, the analysis of [20] concluded that the full information about (massless) excitations is available to any asymptotic observer on null infinity. Our setup is similar in that gravity is dynamical on both the physical and bath branes. Since both branes are $AdS_d$, joined at their common interface which we have been calling the defect above, the role of the asymptotic observer is played by this defect. It is here that gravity shuts off; we are basically studying a universe whose spatial slices are compact, but have a codimension-two wedge on which gravity disappears, in accordance with [41, 42]. In direct analogy with [20], one would want to claim that for a geometry with a single asymptotic region, all information about excitations in the geometry is encoded on this defect, leading to constant entropy. For a geometry with two asymptotic regions like the eternal black string that we study in Section 2.3, the arguments of [20] would imply that all the information is encoded in the union of the two defects. This is, in fact, exactly what wedge holography would indicate: the CFT on the defect encodes the physics of the bulk geometry.

We would like to make two additional comments. First, both the arguments of [20] and of wedge holography would suggest that the fine-grained entropy of the defect as a whole should remain constant even when the bulk geometry is time-dependent. We do not check this claim in our paper since, in the black string geometry, both branes contain black holes that are in thermal equilibrium. But in principle, one could consider a dynamical solution where one starts with a black hole on the physical brane that evaporates into the bath brane. It would be interesting to extend our analysis to this case. Second, even if all the information is available near the defect, it is possible to still obtain a Page curve by dividing the defect itself into two parts as we describe in section 3. This is also consistent with the analysis of [20], which suggested that even when gravity is dynamical it may be possible to find a division of the Hilbert space that yields a Page curve, even though this does not correspond to a division of degrees of freedom into those "outside" and "inside" a black hole.

## 2.3 Empty AdS

We have two options for the bulk geometry; we may either use the equations of Section 2.2 and study the entanglement surfaces in a Poincaré patch of empty $AdS_{d+1}$, or we may use global $AdS_{d+1}$. We start with the former, comparing the result to the field theoretic answer. We then check the global case. Ultimately, all three results will match.

In the Poincaré patch, we need to find solutions of the form (11) obeying (14) on both branes. However, if the endpoints of the RT surface are allowed to vary, then nothing stops them from moving to larger $u$, with the surface expanding to contain the full space. Furthermore, we can see from (12) that the $u^{-2}$ suppression of the metric leads to an area that goes to 0 as $u \to \infty$. Thus, we conclude that, in the Poincaré patch, both $\mathcal{R}$ and $\mathcal{I}$ for the true minimal, extremal surface are reduced to points at infinity, and $S = 0$.

Going to the true field theory dual allows us to make sense of this answer. As in the case of a non-gravitating bath, the two-brane KR setup has a triality of descriptions: the $(d + 1)$-dimensional classical bulk with two branes, a $d$-dimensional gravity theory coupled to a holographic CFT on the branes, and the true field theory dual. In the one-brane case, the purely field theoretic system is a BCFT. However, in the two-brane case, this non-gravitational description is a $(d - 1)$-dimensional CFT living on the defect, as discussed in [41, 42]. This is because

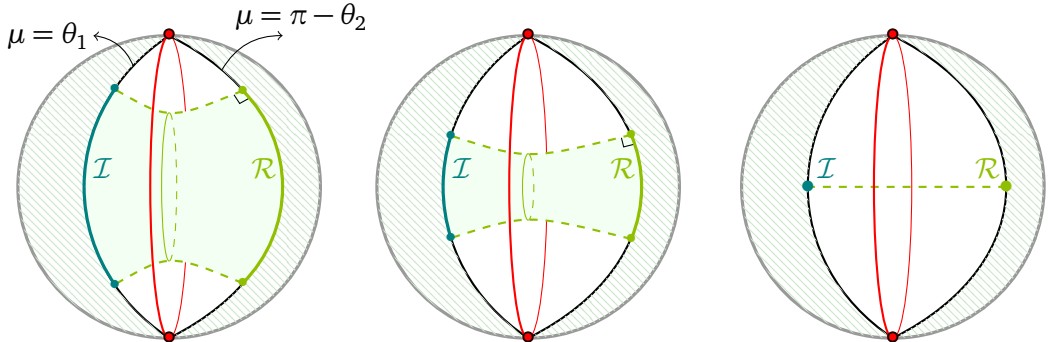

Figure 2: A constant-$t$ slice of global AdS$_{d+1}$ with two branes present. The defect is shown in red, and various candidate extremal surfaces (in order of decreasing area from left to right) are in green. For each surface, $\mathcal{R}$ and $\mathcal{I}$ are respectively the radiation region and island, which end orthogonally on both branes. The minimal, zero-area entanglement surface $r(\mu) = 0$ is shown on the right as a limit, cutting through the middle of the space.

the $\mu$ direction is compactified on an interval, so, technically, the bulk theory is AdS$_d$ times this interval, similarly to how a 10-dimensional string theory on AdS$_5 \times S^5$ has a 4-dimensional dual—not a 9-dimensional dual.

In this CFT, we do not introduce any division of the degrees of freedom into subsets, so the entropy we are calculating is simply the von Neumann entropy of the entire CFT state. As the geometry is the vacuum, the CFT state is pure. Thus, the entropy should be zero. The consistency between this field theoretic answer and what we find in the bulk is another piece of evidence that allowing the endpoints of the RT surface to move so as to satisfy (14) is the correct course of action.

One may wonder if $S = 0$ is also the correct answer when studying the solution in global AdS$_{d+1}$, rather than in the Poincaré patch we have been looking at so far. This means that the field theory is now living on the sphere,[9] so there is now a scale in the problem: the curvature radius of the sphere. Global AdS$_{d+1}$ is also the setting of [16] which established the existence of islands at zero temperature in modified braneworld setups. Our two-brane configuration is depicted in Figure 2.

It is quite straightforward to see that, with our prescription of imposing (14) on both branes, $S = 0$ is still the global answer, even with a scale in the problem. In global AdS$_{d+1}$, the metric (8) is replaced by,

$$ds^2 = \frac{1}{\sin^2 \mu} \left( -\cosh^2 r\, dt^2 + dr^2 + \sinh^2 r\, d\Omega_{d-2}^2 + d\mu^2 \right), \tag{24}$$

where $r$ goes from 0 to $\infty$. Instead of the $d-2$ planar coordinates $\vec{x}$, we now have the coordinates of a $(d-2)$-sphere with volume element $\sim d\Omega_{d-2}$.

The branes are sitting at $\mu = \theta_1, \pi - \theta_2$, as before, with the boundary condition (14) implying $r' = 0$ at both branes. The corresponding area density,

$$\mathcal{A} = \int_{\theta_1}^{\pi - \theta_2} d\mu \frac{(\sinh r)^{d-2}}{(\sin \mu)^{d-1}} \sqrt{1 + r'(\mu)^2}, \tag{25}$$

has a simple solution consistent with the $r' = 0$ boundary conditions,

$$r(\mu) = 0. \tag{26}$$

---

[9]Technically, the defect is homeomorphic to $\mathbb{R} \times S^{d-2}$, with the sphere parameterized by spatial coordinates. In Poincaré coordinates, this defect is homeomorphic to $\mathbb{R}^{d-1}$.

with the corresponding entropy again yielding $S = 0$.

The shrinking of $\mathcal{R}$ and $\mathcal{I}$ is clearer in global coordinates than in the Poincaré patch, in which both regions run off to $u \to \infty$. As one can see in Figure 2, $\mathcal{R}$ and $\mathcal{I}$ are topologically $d$-dimensional balls. In the limiting surface capturing the entanglement entropy, these balls contract to points. The resulting generalized entropy of their union (5) is simply 0.

## 2.4  Black String

To compare to the results of [1–3], we would like to study the system with a black hole on the physical brane. However, a uniform radiation bath in time-independent AdS space is not a solution to Einstein's equations, so we have a choice on how to proceed. We can keep the radiation uniform on the brane and allow for a time-dependent solution. Einstein's equations, which include the jump equations on the brane, force the brane to move in response to the radiation. The position would then be time-dependent and the induced worldvolume metric would be an FRW universe driven by the radiation. The exact solution for a single such brane was worked out long ago in [64].

Alternatively, we can let the radiation in the bath region also collapse into a black hole, in which case we will have a black hole on both branes with matching Hawking temperatures. As long as both branes are above the Hawking-Page transition temperature, the system would be in stable equilibrium. This is exactly the scenario analyzed in [38]. One bulk metric corresponding to this scenario is the $\text{AdS}_{d+1}$ black string metric,

$$ds^2 = \frac{1}{u^2 \sin^2 \mu} \left[ -h(u) dt^2 + \frac{du^2}{h(u)} + d\vec{x}^2 + u^2 d\mu^2 \right], \quad h(u) = 1 - \frac{u^{d-1}}{u_h^{d-1}}. \tag{27}$$

This has the same $\mu$-dependence as for empty $\text{AdS}_{d+1}$ in (8), but this time with planar $\text{AdS}_d$-Schwarzschild black holes, instead of empty $\text{AdS}_d$, on each constant-$\mu$ slice. The corresponding spacetime is sketched in Figure 1b.

Such black string metrics can be afflicted by a Gregory-Laflamme instability [37], but in the case of subcritical branes the instability is absent for large black holes, which includes the case of the planar black hole of interest to us. In fact, it was shown in [38] that the onset of the Gregory-Laflamme instability is the bulk manifestation of the Hawking-Page phase transition on the brane.

We now want to find the entangling surfaces. Since $h(u)$ vanishes at $u_h$, the coordinate system in which we write the metric (27) is not well-suited to analyze the behavior of an RT surface near the horizon. We use the *tortoise-like coordinates*, instead; that is, we redefine the radial coordinate to be,

$$\frac{du}{\sqrt{h(u)}} = dr, \tag{28}$$

so that the $t = 0$ slice of (27) now reads

$$ds_{t=0}^2 = \frac{1}{\sin^2 \mu} \left( \frac{dr^2 + d\vec{x}^2}{u^2} + d\mu^2 \right), \tag{29}$$

where it is understood that $u$ is a function of $r$, obtained from integrating (28). The precise form of $u(r)$ is not needed for this analysis, however.

Parameterizing $r(\mu)$, the corresponding action now reads,

$$\mathcal{A} = \int_{\theta_1}^{\pi - \theta_2} \frac{d\mu}{(u \sin \mu)^{d-1}} \sqrt{u^2 + r'(\mu)^2}. \tag{30}$$

Insisting that the boundary variation of $\mathcal{A}$ vanishes implies the boundary condition $r' = 0$ at both $\mu = \theta_1$ and $\mu = \pi - \theta_2$. Furthermore, in the bulk, the two terms of the Euler-Lagrange equation read,

$$\frac{d}{d\mu}\left(\frac{\partial A}{\partial r'}\right) = \frac{d}{d\mu}\left[\frac{r'}{(u\sin\mu)^{d-1}}\frac{1}{\sqrt{u^2 + r'^2}}\right], \tag{31}$$

$$\frac{\partial A}{\partial r} = \frac{\partial A}{\partial u}\frac{du}{dr} = \frac{\partial A}{\partial u}\sqrt{h(u)}. \tag{32}$$

While it is no longer possible to find the most general solution in closed form, we can readily write down one solution obeying the correct boundary conditions,[10]

$$u(\mu) = u_h. \tag{33}$$

This is the unique solution both obeying the Euler-Lagrange equations and satisfying the orthogonality boundary conditions at both branes, so the black string horizon itself is the single RT surface.

To see that the horizon is indeed the unique solution, we study the equations of motion to rule out other potential candidates. Away from the horizon, we do not need to use tortoise-like coordinates. If we parameterize our surface as $u(\mu)$, and then solve for the equations of motion for general bulk dimension $d + 1$, we obtain,

$$u'' = -(d-2)u\,h(u) + (d-1)u'\cot\mu\left(1 - \frac{\tan\mu}{2}\frac{u'}{u\,h(u)} + \frac{u'^2}{u^2\,h(u)}\right) \\ -\left(\frac{d-5}{2}\right)\frac{u'^2}{u}. \tag{34}$$

If $u'$ vanishes at some angle $\mu = \mu_0$,

$$u''(\mu_0) = -(d-2)u(\mu_0)\left[1 - \frac{u(\mu_0)^{d-1}}{u_h^{d-1}}\right]. \tag{35}$$

In other words, $u''$ is negative unless the surface lies precisely at the horizon. Since $u' = 0$ at the left brane where the RT surface starts, this means that $u' < 0$ just to the right of that brane. Now let's assume that there exists a second brane at which $u'$ also vanishes. By the same argument, $u'' < 0$ here as well. This second brane is supposed to be the endpoint of the RT surface, so we care about the value of $u'$ to its left. Since $u'' < 0$ and $u' = 0$ at the brane, $u' > 0$ just to the left of it.

Thus, $u'$ must change sign between the two branes. There are two cases to consider— either $u'(\mu)$ is continuous or the parameterization breaks down, corresponding to $d\mu/du = 0$. In the former, there exists an angle $\mu_m$ between the two branes at which $u(\mu_m) < u(\theta_1)$, $u'(\mu_m) = 0$, and $u''(\mu_m) \geq 0$. This third condition allows for $u'$ to change sign, but it is in direct contradiction with (35). We deduce that any extremal surface besides the horizon cannot have a continuous derivative.

Thus, for $u'$ to change sign, there must be an angle $\mu_\infty \in (\theta_1, \pi - \theta_2)$ at which,

$$\lim_{\mu\to\mu_\infty^-} u'(\mu) = -\infty, \quad \lim_{\mu\to\mu_\infty^+} u'(\mu) = +\infty. \tag{36}$$

However, such a surface runs along a radial line of constant $\mu$, thus failing to make it "across" the defect and to the other brane. We have thus ruled out surfaces which are not the horizon (33).

---

[10](31) vanishes trivially by $r' = 0$, which is implied when $du/d\mu = 0$. Additionally, (32) vanishes because $\partial A/\partial u$ is finite, but $du/dr = \sqrt{h(u)} = 0$ at $u = u_h$.

The fact that we find our RT surface to be the horizon makes sense physically. Horizons are minimal area surfaces, and the horizon itself obeys the appropriate boundary conditions that we require. Also, the corresponding entropy satisfies,

$$S = \frac{A_{d+1}}{4G_{d+1}} = \frac{A_d}{4G_d}. \tag{37}$$

Here, $A_{d+1}$ stands for the $(d+1)$-dimensional horizon area, $G_{d+1}$ is the $(d+1)$-dimensional Newton constant, and $G_d$ is the $d$-dimensional Newton constant after compactifying $\mu$ on the interval. Note that $G_d$ is not the same as $G_{eff}$, the Newton constant for the massive gravity theory on the physical brane. $G_d$ is instead the Newton constant that governs the dynamics of the true zero mode, which is a superposition of bath and physical brane gravitons. $G_d$ is given by the standard Kaluza-Klein reduction formula for compactification on an interval of finite volume,

$$\frac{1}{G_d} = \frac{1}{G_{d+1}} \int_{\theta_1}^{\pi-\theta_2} \frac{d\mu}{(\sin \mu)^{d-1}}. \tag{38}$$

The same factor of $\int \frac{d\mu}{(\sin \mu)^{d-1}}$ also relates $A_{d+1}$ to $A_d$, so that it is indeed straightforward to confirm that the $(d+1)$-dimensional area in $(d+1)$-dimensional Planck units is the same as the $d$-dimensional area in $d$-dimensional Planck units.

Lastly, from the $(d-1)$-dimensional CFT perspective, the answer is once again easy to understand. Here we are studying a mixed state that is given by the standard thermal density matrix. The entropy is reproduced in the bulk by the horizon area of a static AdS-Schwarzschild black hole [65].

# 3 Left/Right Entanglement at Finite Temperature

We have seen that when gravity is dynamical for both the brane and bath, the conventional entanglement entropy of radiation leads to trivial results. In empty AdS$_{d+1}$ bounded by KR branes, this entanglement entropy is always zero, whereas for a black string this entanglement entropy is always calculated by the horizon area. In neither case do we get a Page curve. In the first case, this is a consequence of empty AdS being in a pure state, whereas for the black string the finite temperature calculation captures the entanglement between the defect system and its thermofield double. This makes it appear that there is no evolution of the system with time. However that is not the case.

In this section we argue that there is time-dependence in some other form of entanglement entropy for finite temperature setups. Even with dynamical gravity on both KR branes, the defect is special in that it is non-gravitating. So we may divide its degrees of freedom into a "left" part and a "right" part. We call the entanglement entropy between these parts the *left/right entanglement entropy*.

We will see that, for a range of $\theta_1$ and $\theta_2$ parameters bounded from above by what we call the *Page angle $\theta_P$* this left/right entanglement entropy can yield a Page curve indicating the evolution of the structure of entanglement of a different type for our system. The physics of the Page angle is driven by the appearance of a *critical angle $\theta_c$*, beyond which some of the potential saddles contributing to the entanglement entropy cease to exist. We will see that this critical angle plays an important role in several different systems and seems to be the critical dividing point for at least two different types of RT surfaces. The apparently universal nature of this critical angle and its implications for the Page curve are not yet fully understood. They might however be a clue to the microscopic origin of the existence of non-trivial saddle points that can lead to a Page curve and in any case might constrain the possibilities. We summarize what we find in Section 3.1 and present the details in those following.

## 3.1 Motivation and Summary

As discussed previously, holography dictates that the $(d+1)$-dimensional classical bulk encodes the quantum effects of the $d$-dimensional description. Equivalently, the same entanglement entropy density can also be calculated in the $(d-1)$-dimensional purely field theoretic dual. In computing the left/right entanglement entropy, we consider potential RT surfaces that start from the defect where the two branes meet. Broadly speaking, there are two classes of candidate surfaces: the Hartman-Maldacena surface [43] connecting the defect to its thermofield double and the *island surfaces* connecting the defect to either brane. The appropriate quantum entangling surface on the brane is chosen via minimization.

In the exterior region of the black string geometry, the Hartman-Maldacena surface goes from the defect to the horizon. However, in the maximally-extended, two-sided black string geometry, there are two defects: the defect displayed in Figure 1b and its thermofield double. The Hartman-Maldacena surface connects these defects going through the bulk Einstein-Rosen bridge, and the portion of the surface located behind the horizon (which has zero area at $t = 0$) increases indefinitely in time due to the growth of the Einstein-Rosen bridge, see Appendix A. Naively, this would introduce an information paradox [43].

Meanwhile, an island surface is defined as connecting the defect to a brane. As the name suggests, these surfaces feature islands, which appear as disconnected regions in the $d$-dimensional computation of the entanglement entropy. In the black string geometry, this class of surfaces saves us from the paradox mentioned above; the island surfaces are located strictly outside of the horizon, so they do *not* grow in time for the eternal black string geometry. The presence of these surfaces also prevents us from strictly interpreting the left/right entanglement entropy as being the entanglement between the entire left and right brane. That they form islands indicates that some degrees of freedom on the left brane are redundant with degrees of freedom on the right brane. However, the division of the defect's internal degrees of freedom into "left" and "right" is well-defined with or without islands. The lack of precise correspondence to the branes themselves is due to the interactions of the left and right degrees of freedom.

A brief summary of our results follows.

We will first demonstrate that in the black string geometry there exists a critical angle $\theta_c$ for the brane location that demarcates regions of consistent RT surfaces. For a single brane with $\theta > \theta_c$, no RT surface obeying the correct boundary conditions (discussed in Section 2.2) and connecting the brane to the defect exists.

However, even though no local extrema exist when minimizing the area as a function of the intersection point of the brane with the surface at angles greater than the critical angle, the minimum value is obtained in an asymptotic limit. In the black string geometry, these surfaces become infinitesimal, but their area is not zero due to the UV divergence of the asymptotic $\text{AdS}_{d+1}$ metric. We call such pseudo-island surfaces *tiny island surfaces*. In the $\theta > \theta_c$ regime, we find that the area of the tiny island surface, while divergent, is infinitely smaller than that of the Hartman-Maldacena surface and so is, in fact, the dominant solution. The validity of these tiny island surfaces can be justified by slightly deforming the system in such a way that the tiny island surface actually has a finite negative area difference with the Hartman-Maldacena surface. We'll provide such a justification both in the single-brane and the two-brane scenario.

On the other hand, for $\theta < \theta_c$, we find a genuine extremal surface ending orthogonally at a finite depth on the brane. We note that the asymptotic tiny island surface mentioned above also exists below the critical angle. However, for small angles, this surface has an area infinitely *larger* than that of Hartman-Maldacena. In fact, we can alternatively define the critical angle as the value of $\theta$ at which the leading divergence in the area difference between Hartman-Maldacena and the tiny island surface flips sign. As such, $\theta_c$ only cares about the near boundary behavior of the metric and should not be sensitive to bulk geometrical properties.

We will explicitly see this play out when comparing the black string to empty $AdS_{d+1}$ (Section 4). So below the critical angle we do not need to consider the tiny island surfaces.

For angles much smaller than the critical angle, the true island surface has an area that is finitely larger than that of the Hartman-Maldacena surface at $t = 0$.[11] This means that the initial entangling surface is the Hartman-Maldacena surface, and there is a phase transition of the entangling surface after some time to the finite-depth island surface. However, at the critical angle, and in a small range of angles below it, the $t = 0$ Hartman-Maldacena surface is already larger than the island surface, and so in this narrow range of angles the entropy is a time-independent constant and the entropy curve is trivial. We call the lower boundary of this region, with constant entropy curve dominated by the island surface, the Page angle.

With the two branes in Figure 1b, we compare the Hartman-Maldacena surface to all possible island surfaces. We find that a Page curve indeed exists for left/right entanglement entropy, but if and only if $\theta_1$ and $\theta_2$ are both below the Page angle. We emphasize that this Page angle, whose existence is driven by the critical angle, is a benchmark for us to get the Page curve.

The critical angle is perhaps the most intriguing aspect of our results and we have already seen that it plays several roles. It is the point where the tiny island surfaces become irrelevant and also where the leading divergence of the area of the finite-depth island surface below the critical angle equals that of the Hartman-Maldacena surface. In addition, $\theta_c$ is the upper bound of the angles for which an island surface can be found at all. This angle also played a role in the analysis of [15]. We will further explore the generality and universality of the critical angle in a later section. Our main interest in the following section will be the physics of the black string geometry, representing a finite temperature state in the defect system.

## 3.2 Black String

Let us now turn to the black string. We will use the exactly known finite temperature solution to understand the evolution of left/right entanglement entropy at finite temperature. This entanglement entropy can be understood as originating from the left and right degrees of freedom on the boundary system, which consists of a $(d-1)$-dimensional thermofield double state.

### 3.2.1 Initial Time Hartman-Maldacena Surface

We choose to use polar Poincaré patch coordinates $u$-$\mu$, where we are confronted with the metric and blackening factor $h(u)$ given by (27). We start by analyzing the Hartman-Maldacena surface, which crosses the event horizon, passes through the Einstein-Rosen bridge, and anchors to the thermofield double defect on the other side of the Einstein-Rosen bridge. As expected for a surface passing through the Einstein-Rosen bridge, the area of this surface increases linearly following a short non-linear growth phase, a result of [43] which we review in Appendix A. Since the RT surface must be anchored to the defect, the Hartman-Maldacena surface must be anchored on both sides of the fully extended black string spacetime.

The general Hartman-Maldacena surface would be parameterized by a curve in the $t$-$u$ plane. However, for the purpose of understanding the Page curve, the first question is whether this surface contributes to the entropy at all. This is determined by the value of the area of this surface at $t = 0$, which is when it is the smallest. In this subsection, we analyze this special $t = 0$ surface, leaving the general time-dependent analysis for Appendix A. Furthermore since we are interesting in numerically comparing the area of this surface to the area of other surfaces, we will specialize to $d = 4$.

---

[11]We technically have two copies of the island surface (one in each of the maximally-extended black string's exterior regions). However, we may also compare the area of one copy to the area of Hartman-Maldacena in one of the exterior regions.

Since the surface lies at a constant angle, the $u(\mu)$ parameterization is inconvenient so we change coordinates to $\mu(u)$. For our initial $t = 0$ slice we have the induced metric on the RT surface

$$d\tilde{s}^2|_{t=0} = \frac{1}{u^2 \sin^2 \mu} \left[ \left( \frac{1}{h(u)} + u^2 \mu'(u)^2 \right) du^2 + d\vec{x}^2 \right]. \tag{39}$$

For convenience we have set $u_h = 1$ which gives $h(u) = 1 - u^3$. Using the induced metric, we get the area functional,

$$\mathcal{A} = \int \frac{du}{(u \sin \mu)^3} \sqrt{\frac{1}{h(u)} + u^2 \mu'(u)^2}, \tag{40}$$

from which we can obtain the equations of motion for the extremal surfaces:

$$\begin{aligned}
\mu''(u) = {} & 3 \cot[\mu(u)] \left[ \frac{1}{u^2(u^3 - 1)} - \mu'(u)^2 \right] \\
& - \left[ \frac{2 + u^3}{2u(u^3 - 1)} \right] \mu'(u) - 2u(u^3 - 1)\mu'(u)^3.
\end{aligned} \tag{41}$$

It is straightforward to see that there is a constant solution, $\mu(u) = \pi/2$, which corresponds to the Hartman-Maldacena surface at $t = 0$. The computation of the area is most easily done in Cartesian coordinates, where we can rewrite the area functional using $\mu'(u) = 0$ and $u \sin \mu = z$. The area is divergent but can be regulated by introducing a small cutoff in the $z$-coordinate.

$$\mathcal{A}_{HM}(0) = \lim_{\epsilon \to 0} \left[ -\frac{1}{2\epsilon^2} + \int_\epsilon^1 \frac{dz}{z^3} \sqrt{\frac{1}{1 - z^3}} \right] = \frac{\sqrt{\pi} \Gamma\left(\frac{1}{3}\right)}{12 \Gamma\left(\frac{5}{6}\right)}. \tag{42}$$

This value will serve as a reference when we compute (and compare) the areas of other surfaces in the following subsections.

### 3.2.2 Island Surfaces

Our next task is to numerically determine those RT surfaces, referred to in this work as island surfaces, which start on the defect and end on one brane where they satisfy the boundary conditions (14). It is important to realize, as we said in the previous section, that we are forced to choose the defect as the point dividing off the radiation region.

These surfaces which start on the defect will end on the brane at some potentially arbitrary anchoring point $u_i$. The boundary conditions enforced by the extremization process, which require the surface trajectory to land orthogonally on the brane, are so restrictive that this point turns out to be determined completely by the strength of gravity on the brane. Each brane then enjoys its own unique anchoring point because the strength of gravity on the brane is uniquely determined by its angle. This point on the brane furthermore divides those surfaces which anchor to the brane and shoot across the defect from those which land on the same side; for this reason we will call this point the *critical anchor*.

For numerical convenience, instead of shooting from the defect to the brane, we enforce the boundary conditions on the brane when determining the solution to the differential equations and shoot to the defect. These conditions are,

$$u(\theta_1) = u_1, \qquad u'(\theta_1) = 0, \tag{43}$$
$$u(\pi - \theta_2) = u_2, \qquad u'(\pi - \theta_2) = 0, \tag{44}$$

where $\theta_1$ and $\theta_2$ are the angles for each brane and $u_i$ ($i = 1, 2$) gives the corresponding anchoring point for the RT surface.

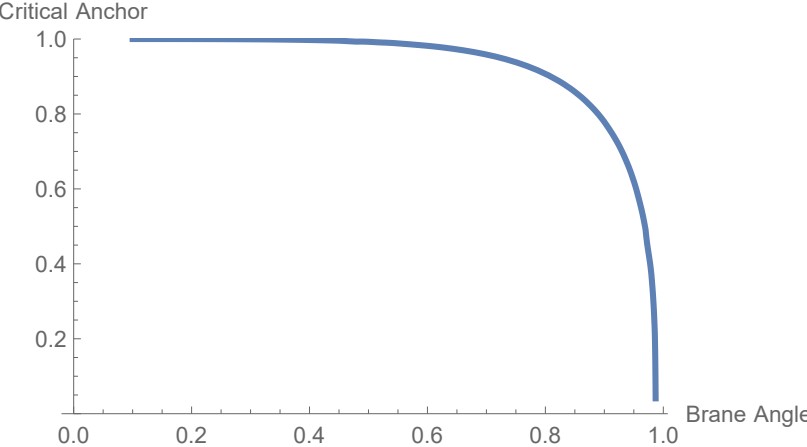

Figure 3: The critical anchor on the brane, given here in four dimensions, is a monotonically decreasing function of the brane angle which vanishes at the critical angle, which in this case is approximately $\theta_c \approx .98687$. The critical anchor tends toward the horizon distance ($u_h = 1$) as $\theta \to 0$ and tends toward zero as $\theta \to \theta_c$.

Before discussing island surfaces in the full two-brane geometry, let us first discuss the case with a single brane, say the physical brane. That is, we first calculate the left/right entanglement entropy in the case of a non-gravitating bath. In this case, one could have fixed a generic radiation region in the bath region. The left/right entanglement entropy here is just the limiting case that the radiation region is chosen to be the entire bath.

By imposing orthogonality, we ensure these surfaces are extremal, but they are not guaranteed to end on the defect for generic $u_i$, so we need to scan through all possible values of $u_i$ to find the critical anchor which allows us to hit the defect. We numerically find those surfaces which hit the defect are only available for branes below a special value for the angle $\theta_c$,

$$0 < \theta_c < \pi/2, \tag{45}$$

which we call the *critical angle*. This angle depends only on the dimension of the space (Figure 8), and the critical anchor is a monotonically decreasing function of the brane angle which vanishes at the critical angle (Figure 3).

The possible island surfaces for a selection of brane angles below the critical angle are illustrated in Figure 4. The critical anchor exhibits interesting limiting behavior as a function of brane angle. Since we cannot shoot to the defect when above the critical angle, we cannot satisfy the boundary conditions. However, every extremal surface starting on the brane above the critical angle makes it over the defect. In fact, the critical anchor slides towards the defect as we approach $\theta_c$ from below. Moreover, the area of the island surface approaches the area of the Hartman-Maldacena surface as one approaches the critical angle. Correspondingly the Page time gets smaller as we approach the critical angle (as illustrated in Figure 6, Section 3.2.4) and vanishes slightly below. Notice that island surfaces near the critical angle lie in the asymptotic region near the defect; this indicates that the value of the critical angle will be tightly constrained by the asymptotic geometry. Another interesting limit is $\theta \to 0$ in which case the critical anchor tends toward the horizon distance $u_h$, the local strength of gravity on the brane vanishes, and the Page time diverges.

The limiting behavior of the critical anchor corresponds directly to the limiting behavior of the island surfaces—one loses the part of the island surfaces outside the horizon as the critical anchor approaches the horizon. In contrast, the island region fills the brane as the

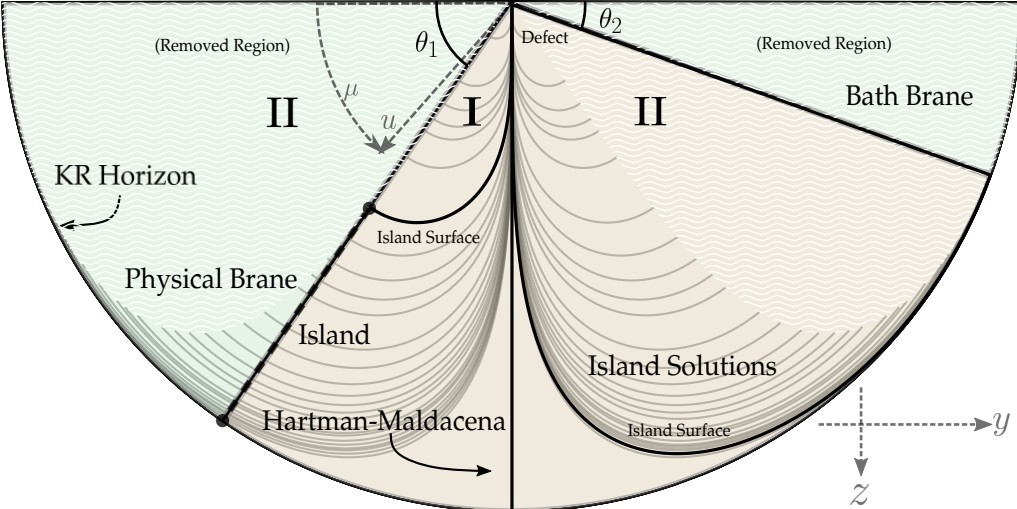

Figure 4: Overview for potential island surfaces within the black string geometry, each of which satisfies the boundary condition for some corresponding brane placed below the critical angle. Each surface ends at its critical anchor, which is where the island begins. One loses the island surfaces as the angle decreases, and the island fills the brane at the critical angle. The area difference between the Hartman-Maldacena surface and the island surface vanishes slightly below the critical angle. While one would need many digits of precision to show this, the island surfaces appear to fill the *island commonwealth*, labelled as region I, without crossing one another. They never make it into the region II, which we call *independent territory*.

critical anchor approaches the defect when the brane hits the critical angle. Phrased in terms of the gravity on the brane, the Page time vanishes continuously as one increases the strength of gravity on the brane; during this process, an island forms and saturates the brane at the critical angle.

### 3.2.3 Above the Critical Angle and Tiny Island Surfaces

For a single brane, the finite island surfaces are lost when the brane lies above the critical angle. We want to show that in this case the area functional is dominated by infinitesimal tiny island surfaces, whose area difference with Hartman-Maldacena (47) is $-\infty$. The left/right entanglement will be a time-independent constant. As mentioned in Section 3.1, these are infinitesimal surfaces obtained as asymptotic limits near the defect that are subdominant when the brane is below the critical angle. However, above the critical angle, the tiny island surfaces are infinitely *smaller* than even the Hartman-Maldacena surface. Coupled with the $\theta \to \theta_c$ behavior of Figure 3, the tiny island surface is the only alternative candidate to Hartman-Maldacena above the critical angle. In fact, we can even extract the critical angle from the behavior of the infinitesimal tiny island surfaces.

If we launch a surface from the defect with a classical trajectory and let it end at a finite $u_i$, but do not impose the Neumann boundary condition $u' = 0$, the area difference (47) becomes a function of $u_i$. We write this difference as $\mathcal{A}(u_i)$. The island surfaces for which $u' = 0$ are genuine local minima of $\mathcal{A}(u_i)$.

One would expect that the properties of these minima depend on the full metric. However, the asymptotic form of $\mathcal{A}(u_i)$ at small $u_i$ is completely determined by the asymptotic form of

the metric. In particular, the leading contribution to $\mathcal{A}$ goes as

$$\mathcal{A} \sim \frac{I}{u_i^2}, \tag{46}$$

with $I > 0$ below the critical angle and $I < 0$ above the critical angle.[12] Although $\mathcal{A}$ is a non-trivial function of $u_i$, its asymptotic form near $u_i = 0$ determines the properties of its minima.

Only when $I > 0$ can we possibly hope to find a single minimum for $\mathcal{A}(u_i)$. Thus, above the critical angle, $\mathcal{A}$ cannot have a single minimum. Any single extremum would have to be a maximum. Thus the characterization of the potential extrema of $\mathcal{A}$ has to change at the same critical angle in any background that is asymptotically AdS. In other words, the critical angle, defined this way, is *universal*.

The case of $I < 0$ corresponds to the dominance of tiny island surfaces as we claimed above. The area difference functional is minimized by sending $u_i$ to zero, where it diverges to negative infinity.

To properly define the tiny island surface in the single-brane case, one can make use of the freedom to change the radiation region when the bath is non-gravitating. Instead of taking the radiation region to be the entire bath, we can instead exclude a small strip of infinitesimal length $\epsilon$ next to the defect from the radiation region. In this case, there exists an island surface whose critical anchor is of order $\epsilon$ yielding a surface which has a large negative, yet finite, area difference with Hartman-Maldacena. In the $\epsilon \to 0$ limit, one recovers the tiny island surface with its negative infinite area difference to Hartman-Maldacena as already noted in [12] and consistent with general AdS/BCFT results [51].

### 3.2.4 Phase Transitions and the Page Curves

The Hartman-Maldacena and island surfaces are both available when working with a single brane below the critical angle. While both have UV-divergent areas, they have the same divergent terms and their area difference is finite with or without regularization.

$$\Delta\mathcal{A}(t) = \mathcal{A}_{IS} - \mathcal{A}_{HM}(t) < \infty. \tag{47}$$

$\mathcal{A}_{IS}$ is the area of the island surface, and $\mathcal{A}_{HM}(t)$ is the area of the Hartman-Maldacena surface at time $t$. $\mathcal{A}_{HM}(t)$ can be said to consist of two terms—one constant in $t$ and another monotonically increasing for $t > 0$.

We display our numerical results for $\Delta\mathcal{A}$ at $t = 0$ in Figure 5. On this initial $t = 0$ time slice the island surfaces are subdominant as long as they attach to a critical anchor on a brane satisfying $\theta \ll \theta_c$. This initial area difference vanishes close to but at a value smaller than the the critical angle, the *Page angle* $\theta_P$. Increasing the angle beyond this point, the island surface dominates already at $t = 0$. Once we reach the critical angle the island saturates the brane. Increasing the angle even further, we lose the island surfaces and the tiny islands take over.

The Page time is determined by this computation of the finite area difference on the initial time slice. For branes above the Page angle, the island dominates from the very beginning and the entropy curve is flat. Even below the Page angle, the island surfaces will eventually dominate. While the island surfaces do not evolve with time even below the Page angle, the Hartman-Maldacena surface eventually grows linearly with time and depletes the area difference after some finite interval called the Page time. At this moment, the island surfaces become dominant, there is a phase transition, and the left/right entanglement entropy saturates. Because the area difference changes sign slightly below the critical angle, the Page time

---

[12]At zero temperature, scale invariance indicates that $\mathcal{A} = I/u_i^2$, so island surfaces exist only at the critical angle, where $I = 0$. We will discuss and exploit this in more detail in Section 4.

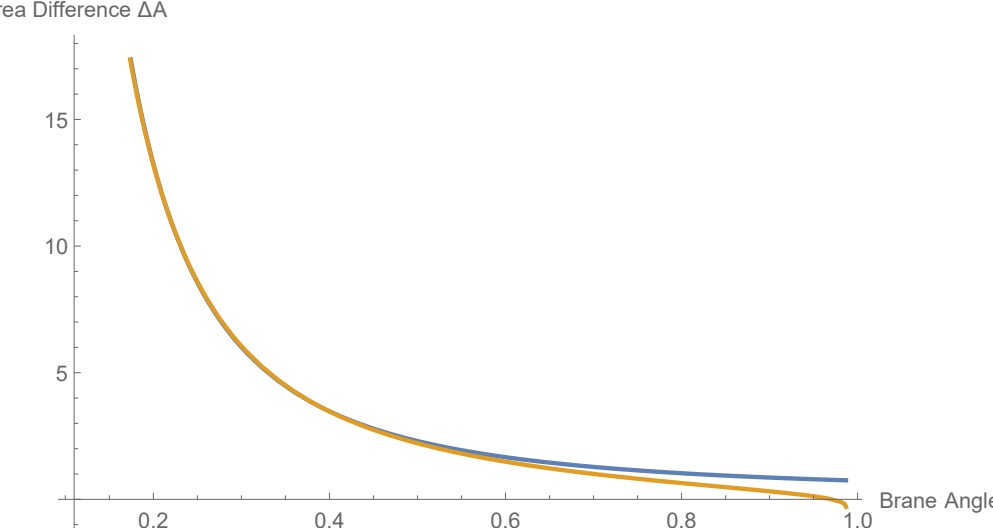

Figure 5: The $t = 0$ area difference (47) between the Hartman-Maldacena surface and the island surface, given as a function of brane angle is in orange, given for the black string at $d = 4$. This should be compared to the behavior of the brane's induced Newton's constant $G_N^{-1} \sim 1/\theta^2$ in blue. As we approach the critical angle the area of the island surface approaches the area of Hartman-Maldacena in empty AdS. This means the difference becomes negative in the region near the critical angle, which for $d = 4$ is approximately $\theta_c \approx 0.98687$.

vanishes there and the islands take over immediately. The resulting single-brane Page curves are sketched for various angles in Figure 6. Note that while Figure 6 displays the entanglement entropy for $t > 0$, this quantity is symmetric about $t = 0$ since $\mathcal{A}_{HM}(-t) = \mathcal{A}_{HM}(t)$.

As we emphasized, our numerics indicate that at the critical angle, the area difference between the island surface and the Hartman-Maldacena surface is actually slightly negative. As a consequence, the area difference vanishes at the Page angle slightly below the critical angle. We can understand this finite value analytically. As noted in Section 3.2.3, at the critical angle the island surface lives entirely in the asymptotic region, and so its properties are equivalent to those of an island surface in empty AdS. This implies, in particular, that the area difference between the island surface and the empty AdS Hartman-Maldacena surface vanishes. Unlike the island surface, the latter actually explores the entire geometry, so we can see that,

$$
\begin{aligned}
\Delta \mathcal{A}^{\theta_c} &= \mathcal{A}_{IS,\mathrm{BS}}^{\theta_c} - \mathcal{A}_{HM,\mathrm{BS}} \\
&= (\mathcal{A}_{IS,\mathrm{BS}}^{\theta_c} - \mathcal{A}_{HM,\mathrm{AdS}}) + (\mathcal{A}_{HM,\mathrm{AdS}} - \mathcal{A}_{HM,\mathrm{BS}}) \\
&= \mathcal{A}_{HM,\mathrm{AdS}} - \mathcal{A}_{HM,\mathrm{BS}} .
\end{aligned}
\tag{48}
$$

Here all areas are calculated at $t = 0$. The BS and AdS subscripts indicate the black string and empty AdS geometries, respectively. The superscript $\theta^c$ reminds us that we do this calculation at the critical angle. The first term on the second line vanishes by the fact that the area difference is 0 at the critical angle in empty AdS. Furthermore, the final result is exactly the negative of that calculated in (42),

$$
\Delta \mathcal{A}^{\theta_c} = -(\mathcal{A}_{\mathrm{HM,BS}} - \mathcal{A}_{\mathrm{HM,AdS}}) = -\mathcal{A}_{HM}(0) = -\frac{\sqrt{\pi}\Gamma\left(\frac{1}{3}\right)}{12\Gamma\left(\frac{5}{6}\right)}.
\tag{49}
$$

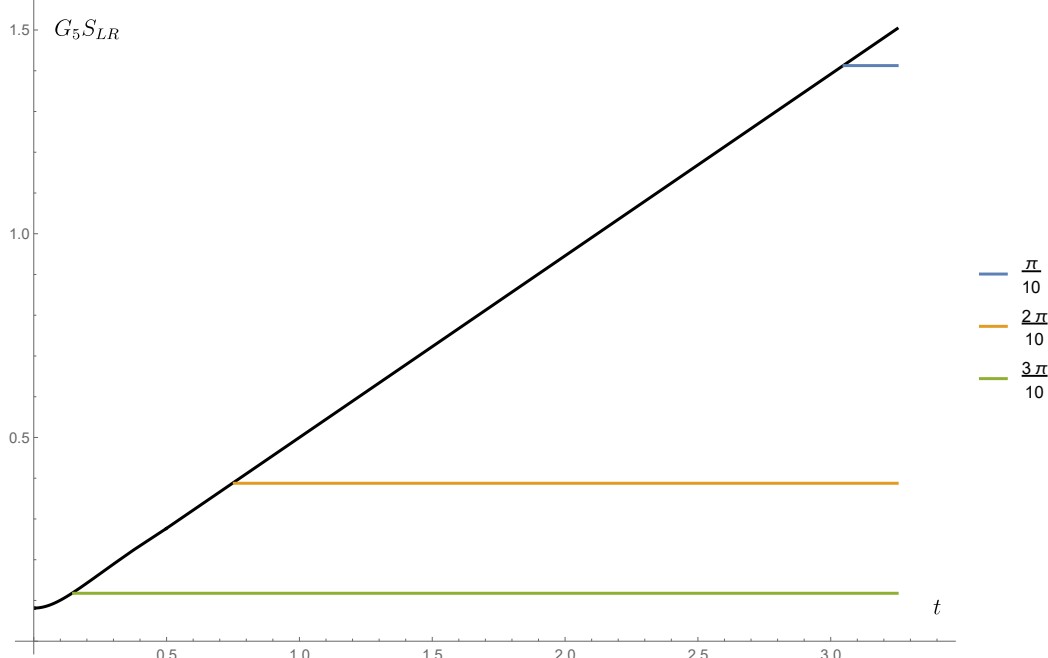

Figure 6: Page curves obtained for sample angles $\theta = \frac{\pi}{10}, \frac{2\pi}{10}, \frac{3\pi}{10}$ starting just below the critical angle $\theta_c \approx 0.98687$. As the brane angle decreases, the Page time goes to infinity. As the brane approaches the critical angle $\theta_c$, the Page time decreases. To obtain the area difference, we integrate numerically up to some small region at which we model the area of the RT surface with a series.

Let us also note the interesting behavior of the area difference, which to a very good approximation is inversely proportional to the induced Newton's constant on the brane.[13] Such behavior would be consistent with the claim that gravity in a KR braneworld is purely induced by matter effects [56].

These results apply as well when we include a weakly gravitating bath if both branes are below the Page angle. While the Hartman-Maldacena surface is unaffected, one of the island surfaces will dominate the other. This means the Page time is completely determined by the strongly gravitating brane, and the addition of the weakly gravitating bath does not change our result. If either brane is above the Page angle, it dominates the entanglement entropy already at $t = 0$, leading to a time-independent entropy curve. Last but not least, if either of the branes is above the critical angle we will once again be dominated by a tiny island surface and in this region we will also have a time-independent left/right entanglement.

Similar to what we did in the one brane case, we can ensure that the tiny island surface in the two-brane case is properly defined by thinking of it as a limiting case. This time we separate the branes at the conformal boundary by a small interval of infinitesimal size $\epsilon$ instead of meeting at a defect. We then take an RT surface ending orthogonally on the brane at a very small $u_i$ and divide the interval into two regions. This regulated situation sports a genuine RT surface whose area is less than that of the Hartman-Maldacena surface, with the difference becoming more and more negative as one shrinks the interval back to zero size.

To summarize, we have two distinct cases. If $\theta_1 < \theta_P$ and $\theta_2 < \theta_P$, the left/right entanglement entropy for the AdS black string follows a Page curve, with the Page time determined by the stronger brane. Otherwise, we have no Page curve, and the left/right entanglement

---

[13]The induced Newton's constant refers to the gravitational coupling of the massive graviton localized to the brane—not the zero mode.

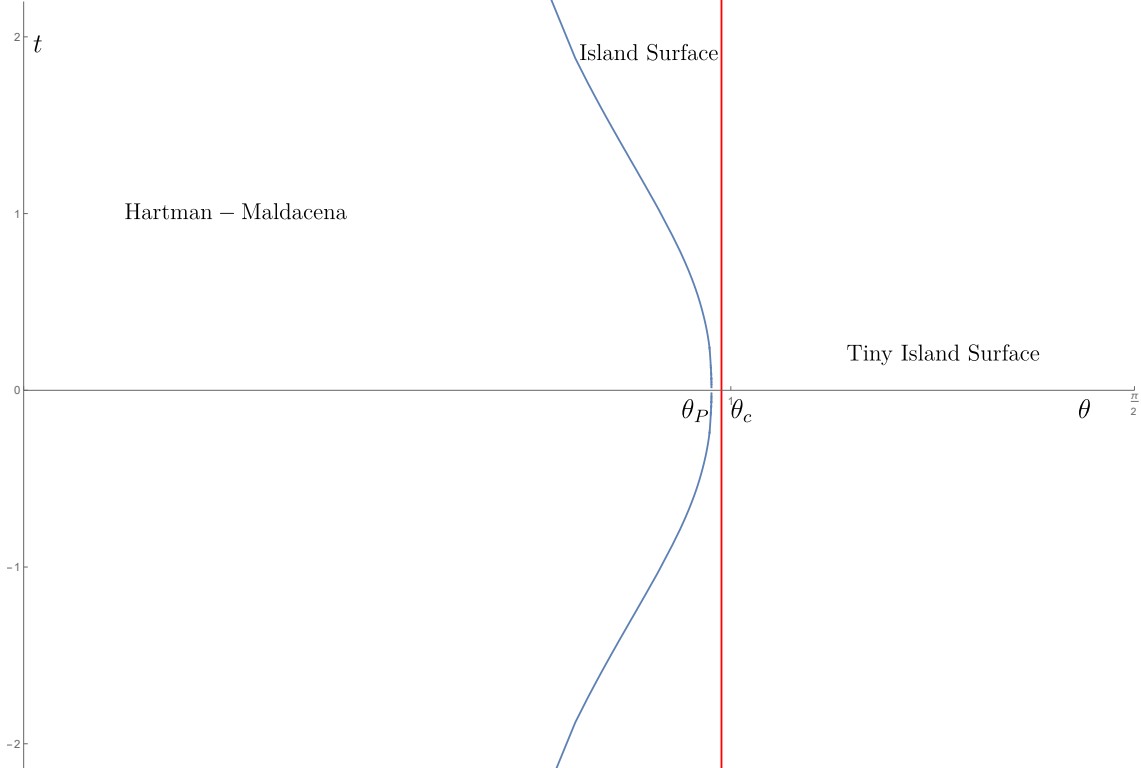

Figure 7: A phase diagram for d=4 showing the dominant minimal surface as a function of time $t$ and the larger brane-angle $\theta$. The transition between the island surface and the tiny island surface always takes place across the critical angle. The transition between the Hartman-Maldacena surface and the island surface is across the blue curve, which intersects $t = 0$ at the Page angle.

entropy is simply constant. Within this second case with at least one of the angles above $\theta_P$, there are two subcases. Either we still have $\theta_1 < \theta_c$ and $\theta_2 < \theta_c$ in which case the left/right entanglement surface is always given by an island surface with finite critical anchor, or one of the angles is above $\theta_c$ and a corresponding tiny island surface dominates.

So there are two important angles. One is the Page angle which is relevant only for the black string. The second is the critical angle, which is the value at which the island surface disappears. This angle can also be defined as the angle at which the island saturates the brane. We can interpret these two angles in terms of two phase transitions. The first at the Page angle is a genuine phase transition from a regime with a time-dependent page curve to one without. The second is more like a spinoidal transition at which the island surfaces disappear. The tiny island surfaces exist at all angles but are important only when their contribution to the entropy difference is negative infinity and hence the smallest area, which is the regime above the critical angle.

The full phase diagram of the system is sketched in Figure 7.

## 4 The Critical Angle

We have seen that the critical angle $\theta_c$ can be used to characterize left/right entanglement surfaces for doubly-holographic models through our numerics for $d = 4$. In particular, this parameter plays an important role in determining whether a given surface can access degrees

of freedom on both sides of the wedge. We have encountered this feature in the black string geometry, but we have also argued that the existence of a critical angle should be determined solely by the asymptotic geometry, where the spacetime increasingly resembles empty AdS (Section 3.2.3). In this section, we first complement the numerical study for the black string with an analytic determination of the critical angle at zero temperature. In the second half of this section, we make some comments about the broader significance of this angle.

## 4.1 The Critical Angle at Zero Temperature

We consider $d > 2$ and study the left/right entanglement surfaces in empty AdS$_{d+1}$ using $y$-$z$ coordinates. We have already found that any extremal surface $y(z)$ satisfies (11),

$$y'(z) = \pm \frac{(z/z_*)^{d-1}}{\sqrt{1-(z/z_*)^{2(d-1)}}},\tag{50}$$

where $z_*$ is the depth of the turnaround point. If we consider a trajectory starting at the defect and with $z_* \to \infty$, we obtain the empty AdS$_{d+1}$ analog of the Hartman-Maldacena surface, for which,

$$y(z) = 0.\tag{51}$$

Alternatively, assuming that the turnaround is finite, one can eliminate the $z_*$ parameter by a change in variables,

$$x = \frac{z}{z_*}, \quad \tilde{y} = \frac{y}{z_*}.\tag{52}$$

The solution then reads,

$$\tilde{y}'(x) = \pm \frac{x^{d-1}}{\sqrt{1-x^{2(d-1)}}}.\tag{53}$$

This equation can be explicitly integrated to obtain the trajectories, which are clearly multi-valued functions $\tilde{y}(x)$ with branch points at $x = 1$ At this stage, we consider the trajectories which reach the KR brane located at $\mu = \theta < \pi/2$.[14] For such a brane, the island surface consists of a $-$ (left-moving) branch anchored to the defect and a $+$ (right-moving) branch starting orthogonally (Section 2.2) to the brane. Starting from the defect, we integrate to obtain an outgoing trajectory,

$$\tilde{y}_{\text{out}}(x) = -\frac{x^d}{d} {}_2F_1\left(\frac{1}{2}, \frac{d}{2(d-1)}; \frac{3d-2}{2(d-1)}; x^{2(d-1)}\right).\tag{54}$$

At the turnaround point $x = 1$, we may use an identity of the hypergeometric function to determine the value of this branch,

$$\tilde{y}_* = -\frac{\sqrt{\pi}}{d} \frac{\Gamma(\frac{3d-2}{2d-2})}{\Gamma(\frac{2d-1}{2d-2})}.\tag{55}$$

Now, we switch to the right-moving branch and integrate until we reach a point on the brane at $x = x_b$ and $\tilde{y} = \tilde{y}_b$, which satisfy the relationship,

$$\tilde{y}_b = -x_b \cot\theta.\tag{56}$$

---

[14]Considering instead $\mu = \pi - \theta$, $\theta < \pi/2$ instead will ultimately yield the same results, but the signs will be flipped on most intermediate steps.

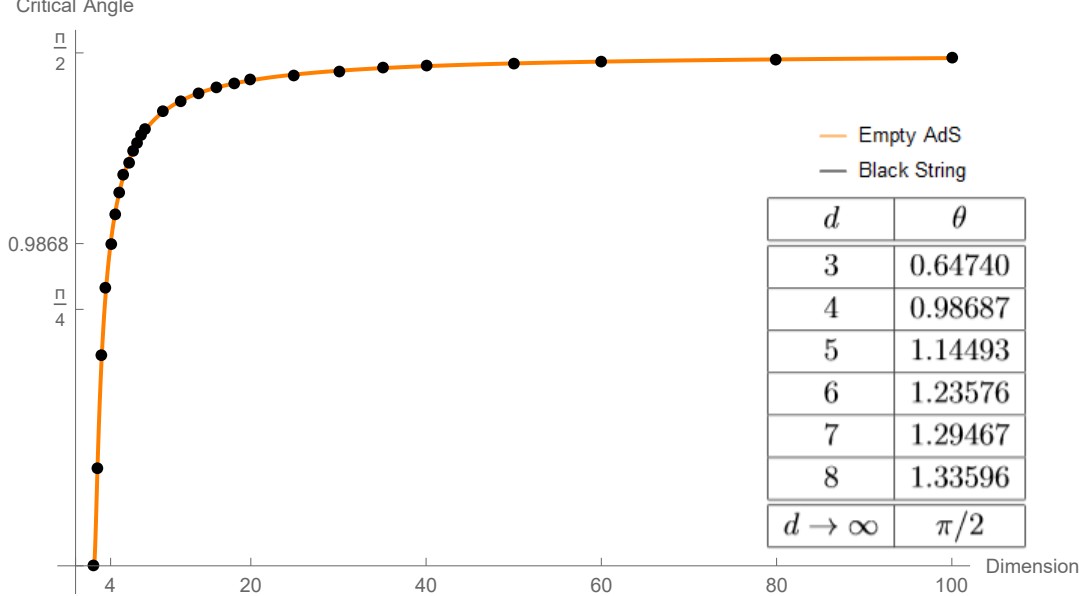

Figure 8: The critical angles in $\text{AdS}_{d+1}$ for various dimensions $d > 2$, analytically continued to real values. The critical angle vanishes at precisely $d = 2$. Both empty AdS and the black string feature the same critical angles; the orange curve was obtained using (59), and the black points were obtained numerically by solving the shooting problem for the black string. The reason for the equivalence is that black string island surfaces near the critical angle are located near the defect, where the metric is well approximated by empty AdS.

For the island surface to be continuous, the value of $\tilde{y}_b$ must also be,

$$
\begin{aligned}
\tilde{y}_b &= \tilde{y}_* + \int_1^{x_b} dx \, \frac{x^{d-1}}{\sqrt{1 - x^{2(d-1)}}} \\
&= \tilde{y}_* + \int_0^1 dx \left[ -\frac{x^{d-1}}{\sqrt{1 - x^{2(d-1)}}} \right] - \int_0^{x_b} dx \left[ -\frac{x^{d-1}}{\sqrt{1 - x^{2(d-1)}}} \right] \\
&= 2\tilde{y}_* - \tilde{y}_{\text{out}}(x_b).
\end{aligned}
\tag{57}
$$

Next, we impose orthogonality at the brane. This reads as (17), which in $\tilde{y}$-$x$ coordinates is,

$$
\tan \theta = \frac{x_b^{d-1}}{\sqrt{1 - x_b^{2(d-1)}}} \iff \sin \theta = x_b^{d-1}.
\tag{58}
$$

Combining our results thus far yields a constraint on $\theta$.

$$
\cot \theta = \frac{2\sqrt{\pi}}{d(\sin \theta)^{1/(d-1)}} \frac{\Gamma(\frac{3d-2}{2d-2})}{\Gamma(\frac{2d-1}{2d-2})} - \frac{\sin \theta}{d} \, {}_2F_1\left( \frac{1}{2}, \frac{d}{2(d-1)}; \frac{3d-2}{2(d-1)}; \sin^2 \theta \right).
\tag{59}
$$

This can be solved numerically, using standard root-finding methods, to compute the values presented in Figure 8 at various dimensions. Observe that the critical angle approaches $\pi/2$ as $d \to \infty$. Additionally in Figure 8, the critical angles computed numerically for $\text{AdS}_{d+1}$ black strings in various dimensions overlap with the plot of real solutions to (59). This emphasizes the universality discussed in Section 3.2.3.

By this method, we have obtained the critical angle analytically by studying island surfaces in empty $\text{AdS}_{d+1}$. In this case, we find that island surfaces exist *only* at the critical angle,

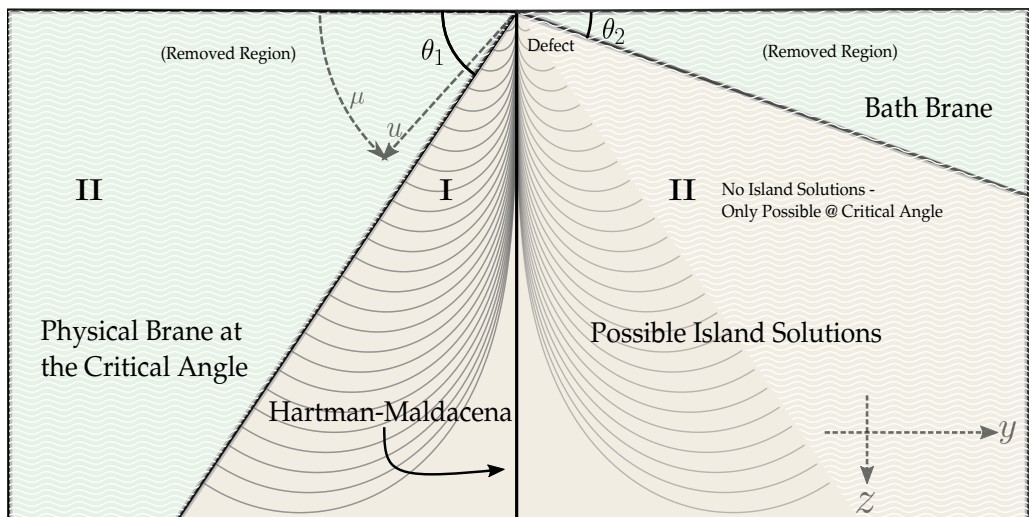

Figure 9: Possible island surfaces in empty AdS$_{d+1}$, which exist only for a brane placed at the critical angle. To summarize, we have two distinct cases. If $\theta_1 < \theta_P$ and $\theta_2 < \theta_P$, the left/right entanglement entropy for the AdS black string follows a Page curve, with the Page time determined by the stronger brane. Otherwise, we have no Page curve, and the left/right entanglement entropy is simply constant.

These are relevant for the black string geometry, since island surfaces near the boundary are increasingly well approximated by empty AdS. The island surfaces completely fill the island commonwealth ($\theta_c < \mu < \pi - \theta_c$), region I, without crossing one another. They also fail to make it into region II, the independent territory ($\mu < \theta_c$ or $\mu > \pi - \theta_c$).

since the orthogonal boundary conditions can be satisfied only at that position for the brane. Candidates for such a surface are depicted in Figure 9.

As observed in Section 3.2.3 the leading divergence in the area difference between Hartman-Maldacena and the tiny island surface vanishes at the critical angle (Figure 5). We now confirm this to also be the case in empty AdS$_{d+1}$.

The area density of the surface (53) is given by $\mathcal{A} = z_*^{2-d} I$, where,

$$I = \lim_{\epsilon \to 0} \left[ -\frac{-1}{(d-2)\epsilon^{d-2}} + \int \frac{dx}{x^{d-1}} \sqrt{1 + \tilde{y}'(x)^2} \right]. \tag{60}$$

To define the integral in (60), we introduce a small cutoff $x = \epsilon > 0$. The integral is performed starting at this cutoff and along the rest of an island surface again hitting a brane at $\theta < \pi/2$. In other words, we will need to integrate along both the outgoing trajectory and the returning trajectory. At the end, the cutoff is taken to zero.

Below we will calculate a regulated version of $I$, using a regularization scheme in which the area of the Hartman-Maldacena area vanishes. It is this regulated $I$ that featured prominently in Section 3.2. We will see that the regulated $I$ vanishes exactly at the critical angle. Furthermore, from the analytic expressions it is apparent that $I$ is positive below and negative above the critical angle, as we already stated in Section 3.2.3.

To determine $I$, we first do the integral along the outgoing trajectory. Denoting this by $I_{\text{out}}$,

we find that,

$$I_{\text{out}}(x) = \lim_{\epsilon \to 0} \left[ -\frac{1}{(d-2)\epsilon^{d-2}} + \int_\epsilon^x \frac{dx}{x^{d-1}} \sqrt{1 + \tilde{y}'(x)^2} \right]$$

$$= -\frac{1}{(d-2)x^{d-2}} {}_2F_1\left( -\frac{d-2}{2(d-1)}, \frac{1}{2}; \frac{d}{2(d-1)}; x^{2(d-1)} \right). \tag{61}$$

This integral is *already regulated* and yields a finite answer for a finite value of $x$. In particular, at the turnaround point,

$$I_{\text{out}}(1) = -\frac{\sqrt{\pi}}{d-2} \frac{\Gamma(\frac{d}{2d-2})}{\Gamma(\frac{1}{2d-2})} = \frac{1}{d-2}\tilde{y}_*, \tag{62}$$

where we have used $\Gamma$-function identities to obtain the second part of the equality above.

If we do the integral in (60) along the outgoing trajectory and the returning trajectory up to $x_b$, we obtain,

$$I(x_b) = 2I_{\text{out}}(1) - I_{\text{out}}(x_b). \tag{63}$$

In our regularization scheme, the area of the Hartman-Maldacena surface is just 0. So, the area difference vanishes if $I(x_b) = 0$, which is satisfied if,

$$2\tilde{y}_* = -\frac{1}{x_b^{d-2}} {}_2F_1\left( -\frac{d-2}{2(d-1)}, \frac{1}{2}; \frac{d}{2(d-1)}; x^{2(d-1)} \right). \tag{64}$$

We rewrite (57) and introduce the boundary condition (58) to alternatively express $2\tilde{y}_*$ as,

$$2\tilde{y}_* = -\frac{\sqrt{1 - x_b^{2(d-1)}}}{x_b^{d-2}} - \frac{x_b^d}{d} {}_2F_1\left( \frac{1}{2}, \frac{d}{2(d-1)}; \frac{3d-2}{2(d-1)}; x_b^{2(d-1)} \right). \tag{65}$$

We conclude that the area difference vanishes if the right-hand sides of (64) and (65) match. We prove that this is indeed the case by some hypergeometric identities. Writing $z = x^{2(d-1)}$ and $a = d/(2d-2)$, we want,

$$\sqrt{1-z} + \left( \frac{2a-1}{2a} \right) z\, {}_2F_1\left( \frac{1}{2}, a; a+1; z \right) - {}_2F_1\left( a-1, \frac{1}{2}; a; z \right) = 0. \tag{66}$$

Once we recognize that the first two entries of ${}_2F_1$ commute and $\sqrt{1-z} = (1-z)\,{}_2F_1(a, 1/2; a; z)$, (66) follows immediately using one of the so-called Gauss contiguous relations.[15] As we mentioned, requiring that the area difference vanishes yields the same critical angle as looking for the existence of solutions.

To summarize, we have two distinct cases. If $\theta_1 < \theta_P$ and $\theta_2 < \theta_P$, the left/right entanglement entropy for the AdS black string follows a Page curve, with the Page time determined by the stronger brane. Otherwise, we have no Page curve, and the left/right entanglement entropy is simply constant.

## 4.2 Significance of the Critical Angle

The critical angle has appeared in several different contexts. It is a consequence of surfaces being dominated by the asymptotic region of AdS space so that the geometry away from the boundary plays little or no role. Here we see how this property manifests itself in the existence of islands in several different contexts; gravitating and non-gravitating bath as well as black string or empty AdS. Some of the points following might seem redundant but here we clarify the different role it plays in these distinct systems.

---

[15]See identity 15.5.16 in [66].

I **Empty AdS with a gravitating bath.**

    (a) Island surfaces ending at the defect exist only at the critical angle.

    (b) The area difference between the island surface and the Hartman-Maldacena surface vanishes at the critical angle.

    (c) The island surface at the critical angle can be taken to end anywhere on the brane by scale invariance. In particular, there exists a choice of island surface for which the island saturates the brane.

II **Empty AdS with a non-gravitating bath.**

    (a) Island surfaces that cross the defect and end in the bath exist only above the critical angle. There is an extremal surface connecting every point on any brane above the critical angle to the bath.

    (b) If one fixes one endpoint of the surface in the bath, and approaches the critical angle from above, the other endpoint of the surface on the brane runs off to the Poincaré horizon. This point was uncovered independently in [15].

    (c) It is only for the critical angle that surfaces starting at a finite anchor point on the physical brane can be found to hit the defect in empty $\mathrm{AdS}_{d+1}$.

III **Black string with a gravitating bath.**

    (a) For the black string, island surfaces ending at the defect exist only below the critical angle.

    (b) The critical angle is where the leading divergence in the area difference between the island surface and the Hartman-Maldacena surface vanishes. This divergence changes sign above the critical angle, which is the region where the tiny island surfaces dominate the entanglement entropy calculation.

    (c) The Page time vanishes slightly below the critical angle. This is because the island surface at the critical angle has the same area as the Hartman-Maldacena surface at zero temperature, which is slightly smaller than its value at finite temperature (42).

IV. **Black string with a non-gravitating bath**

    (a) Below the critical angle, only those surfaces that are anchored beyond a critical anchor on the brane reach the bath. The critical anchor is an asymptotically decreasing function of the angle which decreases from the horizon and vanishes at the critical angle.

The universality of this angle for a given dimension suggests a non-trivial phase transition occurring at the critical angle in many different aspects of KR brane-worlds. For now, we have focused on the implications of our result for the Page curve and leave further study concerning universality of the critical angle to future work, with some speculations in the concluding section.

## 5 Conclusion

In this work, we have analyzed the Page curve for black holes in AdS coupled to a gravitating bath using KR branes as a tool. Our main findings were that gravity in the bath leads to

several major modifications of the Page curve story. While coupling to a non-gravitating bath via boundary conditions implies massive gravitons, extending beyond this analysis to include a gravitating bath means that we are actually studying a theory with a massless graviton. We have shown how gravity in the bath completely changes the behavior of the Page curve for conventional entanglement entropy. If one asks the same questions as has been done in previous work with a non-gravitating bath, *i.e.* computing the entanglement entropy of some region in the bath identified as the radiation region, the entropy is simply constant. This is consistent with the findings of [20] for the entanglement entropy of black holes in asymptotic flat space.

However, we have identified a different quantity, what we call the left/right entanglement entropy, which still yields a Page curve for a fixed range of brane tensions. In the analysis of this left/right entanglement, a crucial role is played by a critical angle. As a function of dimension, this angle goes to zero for $d = 2$ and to $\frac{\pi}{2}$ for infinitely large dimension, and is approximately 0.98687 for $d = 4$. Well below the the critical angle, we find a Page curve with initial entropy growth terminated by island formation. At an angle slightly below the critical angle, the Page curve disappears.

In all previous work on massive gravitons in KR brane worlds, the physics has always been found to depend smoothly on the angle and thus the graviton mass. Here there are two dominant effects as we increase the angle: the UV cutoff decreases and the graviton mass increases, with the latter due to quantum CFT effects. For both of these phenomena, the transition has been gradual. However by studying the entanglement entropy, we have found the first quantity (near the critical angle) which displays a sharp phase transition at a fixed value of the mass. Gravity appears to be very different when we cross the critical angle, and it will be very interesting to better understand this phase transition in future work.

We now briefly speculate on this point. One possible answer may be found by looking at the dual field theory description. Here, there are two competing quantities: the total Hilbert space dimension and the initial entanglement entropy at $t = 0$. If the Hilbert space dimension is small enough such that the initial entanglement entropy saturates its maximal value, then there is no room for the entanglement entropy to grow. Hence, we would have a constant curve for all time. On the other hand, if the Hilbert space dimension is large enough such that the initial entanglement entropy is relatively small, then there is still room for it to grow. Therefore, due to the fast scrambling dynamics of holographic theories [44, 45, 67], we get a Page curve in which the entanglement entropy initially grows, then saturates its maximal bound.

If we consider AdS$_d$ gravity coupled to a CFT, both living on the KR branes, it is well known that the boundary conditions encoded by the angle lead to an angle-dependent UV cutoff and graviton mass [34]. At generic angles, the mass is of order the inverse AdS$_d$ curvature radius on the brane, but at small angles specifically, the mass is parametrically lighter. Therefore the UV cutoff decreases whereas the mass increases as we go to larger angle, perhaps decreasing the size of the Hilbert space to below the required amount for a Page curve transition. It is natural to think that the tension of the brane encodes the Hilbert space dimension of the $(d-1)$-dimensional conformal field theory.[16] Hence, since a Page curve requires a sufficiently large Hilbert space dimension, this implies the existence of a critical angle $\theta_c$ as described above. To see the Page curve for left/right entanglement entropy in the two-brane configuration, both the left system and right systems need to be large enough. This might explain why we would require both $\theta_1 < \theta_c$ and $\theta_2 < \theta_c$.

One crucial take-home lesson from our work is that the coupling to a non-gravitating bath really is an important ingredient in the recent Page curve calculations. The bath is not just a

---

[16]A straightforward example is the tensionless probe brane $\theta = \pi/2$ studied in [12], where the $(d-1)$-dimensional dual is empty.

spectator—it influences the physics. Once we make the bath itself gravitating, although we recover a massless graviton, the original Page curve vanishes. Nevertheless, different quantities can be defined that still sometimes lead to a Page curve, even though its interpretation is unclear. This Page curve is not *the* Page curve of an evaporating black hole. Rather, it indicates the time-dependence of the entanglement entropy between two parts of a holographic system, even though the system as a whole is in thermal equilibrium.

# Acknowledgments

We are grateful to Jan de Boer, Raphael Bousso, Shira Chapman, Roberto Emparan, Severin Lüst, Juan Maldacena, Shiraz Minwalla, Rashmish Mishra, Rob Myers, Yasunori Nomura, Kyriakos Papadodimas, Hao-Yu Sun, Zixia Wei, Liz Wildenhain, Edward Witten and Yoav Zigdon for helpful discussions. We thank Severin Luest and Rashmish Mishra for reading the manuscript. We would like to thank the organizers of the CERN workshop "Island Hopping 2020: from wormholes to averages," for organizing a stimulating virtual meeting. The work of AK was supported, in part, by the U.S. Department of Energy under Grant No. DE-SC0011637 and by a grant from the Simons Foundation (Grant 651440, AK). The work of SR was partially supported by a Swarnajayanti fellowship, DST/SJF/PSA-02/2016-17 from the Department of Science and Technology (India). SS was supported by National Science Foundation (NSF) Grants No. PHY-1820712 and PHY–1914679. The work of LR is supported by NSF grants PHY-1620806 and PHY-1915071, the Chau Foundation HS Chau postdoc award, the Kavli Foundation grant "Kavli Dream Team", and the Moore Foundation Award 8342. HG is very grateful to his parents and recommenders.

# A  Hartman-Maldacena Area in the Black String

We review the analysis of [43] and generalize it slightly to suit our work. We are interested in Hartman-Maldacena minimal surfaces in the $AdS_{d+1}$ black string geometry bounded by two branes. The branes are $d$-dimensional, and the metric is given by (27).

Let us consider a surface that travels at a constant value of $\mu = \frac{\pi}{2}$ from $t = t_{\text{diff}}$ at the original asymptotic boundary to the same value of $t$ at the asymptotic boundary in the thermofield double. This surface can be parameterized by a function $u(t)$. Defining,

$$L \equiv \frac{1}{u^{d-1}} \sqrt{-h(u) + \frac{\dot{u}^2}{h(u)}}, \tag{67}$$

the area that we need to compute is given by the extremum of the action,

$$\mathcal{A} = \frac{1}{(\sin \mu)^{d-1}} \int dt \, L. \tag{68}$$

Since the Lagrangian is not explicitly dependent on $t$, we obtain a quantity that is conserved along any solution of the equations of motion. We call this $C$.

$$C = \dot{u} \frac{\partial L}{\partial \dot{u}} - L = \frac{h(u)u^{1-d}}{\sqrt{\frac{\dot{u}^2}{h(u)} - h(u)}}. \tag{69}$$

We need to be careful about the *sign* of $\dot{u}$, as it changes across the horizon. Solving the

equation above tells us,

$$\dot{u} = \begin{cases} +\dfrac{h(u)}{C}\sqrt{C^2 + u^{2(1-d)}h(u)} & \text{if } u < u_h, \\[3mm] -\dfrac{h(u)}{C}\sqrt{C^2 + u^{2(1-d)}h(u)} & \text{if } u > u_h. \end{cases} \tag{70}$$

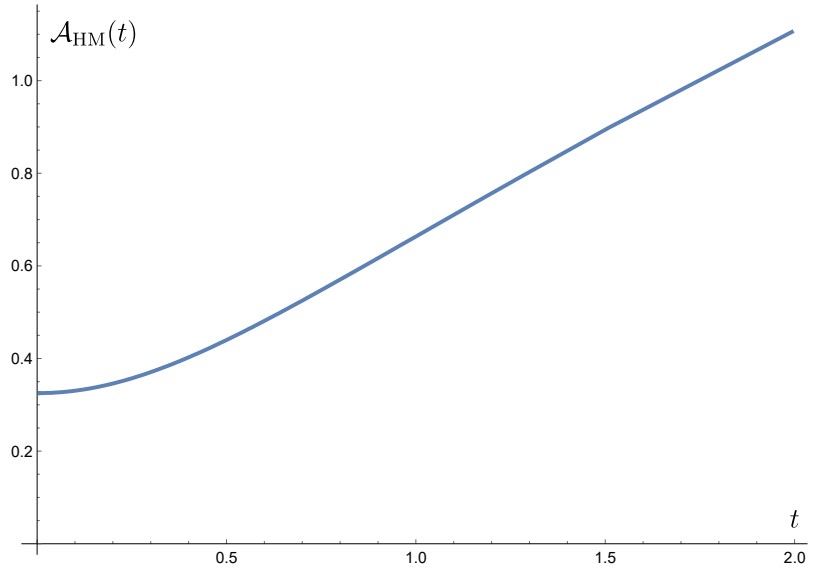

Figure 10: The minimum area versus time for the black string geometry with $u_h = 1$ and $d = 4$.

We can use the symmetry of the system to integrate the area up to the point where $\dot{u} = 0$. This happens at a value $u_s$ that solves,

$$C^2 = -h(u_s)u_s^{2(1-d)}. \tag{71}$$

Note that this point is inside the horizon and so $h(u_s) < 0$ at this point. The minimal trajectory $u(t)$ is symmetric about this point. Recall that the $t$ variable we have displayed here is the *Schwarzschild time*. We are looking for a surface that reaches the thermofield double CFT boundary at the same value of the *CFT time*, which is related to the Schwarzschild time by a minus sign.

We cannot choose the constant $C$ arbitrarily. For example, denote the maximum value of the right-hand side of (71) for $u \in [u_h, \infty)$ by $C^2_{\max}$. If we take $C > C_{\max}$, then (71) ceases to have a solution.

We can also fix $\sin \mu = 1$. The minimal area, which is denoted by $\mathcal{A}_{\text{HM}}$ in section 3.2.4, is then just given by,

$$\mathcal{A}_{HM}(t_{\text{diff}}) = 2 \lim_{\epsilon \to 0} \left[ \frac{-1}{(d-2)\epsilon^{d-2}} + \int_{\epsilon}^{u_s} \frac{du}{|\dot{u}|} \frac{1}{u^{d-1}} \sqrt{-h(u) + \frac{\dot{u}^2}{h(u)}} \right], \tag{72}$$

where we have integrated up to $u_s$ and multiplied by 2 using the symmetry of the minimal surface. We have also removed a universal divergent piece near the boundary. In this integral, $\dot{u}$ is substituted using (70). The time-difference between $u = 0$ and $u = u_s$ is given by,

$$t_{\text{diff}} = \lim_{\delta \to 0} \left( \int_0^{u_h - \delta} \frac{du}{\dot{u}} + \int_{u_h + \delta}^{u_s} \frac{du}{\dot{u}} \right). \tag{73}$$

We can increase the value of $t_{\text{diff}}$ by increasing the value of $C$. As we take $C \to C_{\text{max}}$, both the minimal area and the time-difference start to increase in an unbounded manner. Asymptotically, we find that $\mathcal{A}_{\text{HM}}(t_{\text{diff}})$ varies linearly with $t_{\text{diff}}$, as shown in Figure 10 for $d = 4$.

This area can be interpreted as one possible contribution to the left/right entanglement entropy. However, the unbounded linear growth above potentially leads to a paradox since the entanglement entropy cannot grow without limit. The resolution to this paradox—as explained in section 3.2.4—is that whenever these surfaces contribute to the entanglement entropy at $t_{\text{diff}} = 0$, at late times this growth is cut off when the area exceeds the area of the island surface of section 3.2.2.

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
