# Peer review of "Information Transfer with a Gravitating Bath"

_SciPost Physics, doi:SciPost Phys. 10, 103 (2021)_

## Round 1 · Referee Report · Edgar Shaghoulian · 2021-3-15

Strengths
The paper's primary strength is a concrete brane model where the radiation bath and the gravitational region are both gravitating, within which computations can still be performed. This model in particular has a surface (the interface between the two regions) that is non-gravitating and that allows nontrivial entanglement entropies to be defined and computed.
Weaknesses
The primary weakness is a philosophical one: the theme of the paper is that one is not allowed to shut off gravity in realistic scenarios like our universe where there is gravity everywhere, so they would like to consider that case. However, the nontrivial results of their paper are precisely in a situation where gravity is shut off on the interface between their two gravitating regions.
Report
This is a nice paper with novel calculations which merits publication in SciPost Physics. I would again like to highlight for the reader the non-gravitating interface on which entanglement entropy computations are performed as an interesting new feature.
Requested changes
I would suggest the authors cite Penington's 1905.08255 when introducing the "island formula" in equation (2.1).

---

## Round 1 · Referee Report · Anonymous · 2021-4-2

Strengths
1. The paper is discussing an issue that is very topical, namely to what extend the island formula can be applied to a situation where the bath region that collects the Hawking radiation has dynamical gravity. The paper is therefore a very useful addition to the discussion.
2. The paper describes concrete and novel set ups and comes with a very thorough analysis.
Weaknesses
I have not identified any specific weaknesses of the paper.
Report
The paper is discussing the really interesting question as to whether all the progress made on understanding and solving the information loss paradox using replica wormholes leading to the island formula for the entropy of the radiation can be applied to realistic black holes where the bath region far from black hole still has dynamical gravity. This gets to the heart of some of the deepest questions of quantum gravity: to what extent one can define local quantities like the entropy of a subregion in a QFT in a situation where gravity is dynamical. The paper is adding something useful to the on-going debate, is well written and I cannot see any obvious errors. The conclusion of the paper seems to back up the point of view that there is an essential difference between the cases having no gravity in the bath and cases with dynamical gravity. I can recommend its publication.

---

## Round 2 · Author Response

We would like to thank the referees for their reports. We have inserted the citation pointed out by Dr. Shaghoulian. We have also inserted some additional references and fixed some small typos.

---

## Editorial Decision

published